# Design, Synthesis and Assay of Novel Methylxanthine–Alkynylmethylamine Derivatives as Acetylcholinesterase Inhibitors

**DOI:** 10.3390/molecules27248787

**Published:** 2022-12-11

**Authors:** Danila V. Reshetnikov, Igor D. Ivanov, Dmitry S. Baev, Tatyana V. Rybalova, Evgenii S. Mozhaitsev, Sergey S. Patrushev, Valentin A. Vavilin, Tatyana G. Tolstikova, Elvira E. Shults

**Affiliations:** 1N.N. Vorozhtsov Novosibirsk Institute of Organic Chemistry, Siberian Branch of the Russian Academy of Sciences, Lavrentyev Ave, 9, 630090 Novosibirsk, Russia; 2The Federal Research Center Institute of Molecular Biology and Biophysics, Timakov Str., 2/12, 630117 Novosibirsk, Russia; 3Novosibirsk State University, Pirogova Str., 1, 630090 Novosibirsk, Russia

**Keywords:** methylxanthines, caffeine, theophylline, theobromine, A^3^-coupling reactions, acetylcholinesterase inhibitor, molecular docking study

## Abstract

Xanthine derivatives have been a great area of interest for the development of potent bioactive agents. Thirty-eight methylxanthine derivatives as acetylcholinesterase inhibitors (AChE) were designed and synthesized. Suzuki–Miyaura cross-coupling reactions of 8-chlorocaffeine with aryl(hetaryl)boronic acids, the CuAAC reaction of 8-ethynylcaffeine with several azides, and the copper(I) catalyzed one-pot three-component reaction (A^3^-coupling) of 8-ethynylcaffeine, 1-(prop-2-ynyl)-, or 7-(prop-2-ynyl)-dimethylxanthines with formaldehyde and secondary amines were the main approaches for the synthesis of substituted methylxanthine derivatives (yield 53–96%). The bioactivity of all new compounds was evaluated by Ellman’s method, and the results showed that most of the synthesized compounds displayed good and moderate acetylcholinesterase (AChE) inhibitory activities in vitro. The structure-activity relationships were also discussed. The data revealed that compounds **53**, **59**, **65**, **66**, and **69** exhibited the most potent inhibitory activity against AChE with IC_50_ of 0.25, 0.552, 0.089, 0.746, and 0.121 μM, respectively. The binding conformation and simultaneous interaction modes were further clarified by molecular docking studies.

## 1. Introduction

The natural methylxanthines caffeine **1**, theobromine **2**, and theophylline **3** (Figure 1) are widely used in traditional medicines and also in the field of biological and medicinal chemistry [1]. Methylxanthines exert anti-inflammatory, antioxidative, and neuroprotective effects, so their consumption appears to be useful in the prevention of neurodegeneration [2]. Normally, four different mechanisms are proposed to mediate pharmacological methylxanthine activity at the cellular level: antagonism of adenosine receptors, modulation of GABA receptor action, acetylcholinesterase inhibition, and regulation of intracellular calcium levels [3,4,5,6,7]. In the past years, the impact of methylxanthines in neurodegenerative diseases has been extensively studied, and several new aspects have been elucidated [2,8,9]. Caffeine proved to be a relatively potent non-competitive inhibitor of acetylcholinesterase AChE [2,10,11,12,13,14]. Caffeine acts as a reversible inhibitor of acetylcholinesterase. It binds outside the catalytic site of cholinesterase but in its proximity to the so called anionic site and inhibition, it is of a non-competitive type in the acetylcholinesterase case while caffeine exerts minimal affinity toward the second type of cholinesterase, butyrylcholinesterase, found in the blood plasma and serum [14]. Further studies have confirmed that caffeine **1** is an agonist of nAChRs [15,16]. All the reviewed data on the activity of natural methylxanthines stimulated the development of methods for modifications of the substituent in positions N-1, N-7, and C-8 of the xanthine core to find new compounds with selective action and pharmacological interest.

Broad modifications of the substituent in position 8 of the xanthine core structure have resulted in high potency and selectivity for the A_2A_ [17] and A_2B_ [18] adenosine receptors. The synthesis and biological evaluation of xanthines with an *N*-benzyl-(piperidine or pyrrolidine) substituent and a methoxymethyl linker in N-7, C-8, or N-9 positions revealed dual inhibitors of acetylcholinesterase and butyrylcholinesterase [19,20]. Modification of caffeine at N-1 with a methylpentanoate moiety lead to pentoxyphylline, which exhibited selective inhibition of *h*AChE with no inhibition of *h*BuChE (IC_50_ > 50 μM) relative to the reference agent donepezil [21]. Connecting theophylline **3** with pyrrolidine substituent through a methylene chain of different lengths (three to seven carbon atoms) gave new 7-substituted theophylline derivatives that inhibited the AChE, of which the compound with the longest methylene chain showed the strongest effect. Electrophysiological studies showed that all compounds behave as agonists of the muscle and the neuronal α7 nAChR with greater potency than caffeine **1** [22].

Propargylamine as a versatile functional moiety has been reported to be useful for the design of CNS-targeting multi-target directed ligands for neurodegenerative disorders [23,24,25,26,27,28,29,30,31,32,33]. As represented by such modified preparation as ladostigil, donepezil-based hybrid ASS234, memantine-propargylamine, tacrine-selegiline, and chromene-propargylamine hybrids [34,35,36,37,38], the propargylamine fragment has been identified to be essential for the beneficial effects of these compounds.

In this work, we report the synthesis of new xanthine derivatives based on transformations of natural methylxanthines **1**–**3**: 8-aryl(hetaryl) substituted, 8-(1,2,3-triazolyl)substituted, and substituted at the nitrogen in the side chain 8-propargylamino-based caffeine derivatives, 1-(4-aminobut-2-ynyl)theobromines, and 7-(4-aminobut-2-ynyl)theophyllines (Figure 2) intended as AChE inhibitors.

All derivatives were evaluated for AChE inhibition. To better understand the key structural requirements for effective receptor-ligand interactions in parallel with the biological evaluation, docking studies have also been performed. 

## 2. Results and Discussion

### 2.1. Chemistry

8-Chlorocaffeine **4** and 8-bromocaffeine **5**, the key compounds in the synthesis of all the desired 8-substituted xanthine derivatives, were prepared by reacting caffeine **1** with N-chlorosuccinimide or N-bromosuccinimide according to a general method [39] with minor modifications in the synthetic procedure [40]. In accordance with [41], the Suzuki cross-coupling reaction of 8-bromocaffeine **5** with benzeneboronic acid **6** in the presence of Pd(PPh_3_)_4_ (5 mol %) as the catalyst and K_3_PO_4_ (2 equiv.) as the base in DMF (110 °C, 6 days) led to 1,3,7-trimethyl-8-phenyl-3,7-dihydro-1*H*-purine-2,6-dione **7** isolated in 63% yield. Currently, as an efficient way to improve the rate of the cross-coupling reaction, the addition of tetraalkylammonium salts as additives, given their ability to generate a stable form of the catalyst [42], has been considered. Additionally, microwave-assisted organic synthesis (MAOS) has received increasing attention as a valuable alternative to conventional heating to speed chemical reactions (simplicity in operation, shortened reaction times, giving more pure products) [43]. We found that the reaction of 8-chlorocaffeine **4** with benzeneboronic acid **6** (1.2 equiv) in the presence of Pd(PPh_3_)_4_ (5 mol %) as the catalyst, K_2_CO_3_ (3 equiv.) as the base, and tetrabutylammonium bromide (1 equiv.) as the additive in toluene-water (1:1, *v*/*v*) under microwave irradiation at 130 °C for 1.5 h proceeds well and led to compound **7** isolated in 70% yield (Figure 1). Carrying out the reaction by using an excess of triethylamine as a base was found to be ineffective. The coupling reaction of 8-chlorocaffeine **4** with arylboronic acids **8**–**13** under the mentioned conditions led to 8-arylcaffeine derivatives **14**–**19** in the yield **of** 47–87%. The yield of 8-arylcaffeines was dependent on the nature of arylboronic acid. The main decrease in the isolation yield to 27% was observed in the reaction of **4** with 2,5-dimetoxyphenylboronic acid **12**. The cross-coupling reaction of compound **4** with furan-3-boronic acid **20** or indole-5-boronic acid **21** under the mentioned conditions afforded the desired 8-furyl- or 8-indolylsubstituted caffeine derivatives **22** and **23**, respectively (Figure 1). 

Based on the Sonogashira cross-coupling reaction, a synthetic route was designed for the preparation of 8-ethynylcaffeine **24**. We found that the reaction of 8-bromocaffeine **5** with (trimethylsilyl)acetylene **25** in the presence of Pd(PPh_3_)_4_ (5 mol %) and CuI (5 mol%) as catalysts and Et_3_N (1.5 equiv.) as the base in toluene proceeds smoothly by heating to 95 °C for 6 h (TLC) with the formation of the 8-(3-trimethyl- silyl)ethynylcaffeine **26** (yield 93%). Compound **26** was further subjected to desilylation to furnish the terminal acetylene **24** (yield 88%). This approach to preparing the targeted xanthine derivative was found to be more economical and convenient compared to the previously described procedure [44]. With 8-ethynylcaffeine **24** in hand, we studied the CuAAC reaction with several organic azides **27a**–**f**. Negligible activity was observed when 8-ethynylcaffeine **24** reacted with *p*-tolylazide **27a** in the presence of sodium ascorbate (1.5 equiv.) and copper(II) sulfate (0.1 equiv.) in a mixture of water and chloroform (1v:5v). Low yield of the triazole **32** was obtained after heating to 50 °C for 24 h. Interestingly, we found that the use of CuI (0.1 equiv.) in conjunction with DIPEA (3.5 equiv.) in MeCN dramatically improved the catalytic efficiency, and the reactions were reasonably selective for 8-((1-*p*-tolyl)-1,2,3-triazol-4-yl)caffeine **28** (yield 84%). In this condition, the CuAAC reaction of alkyne **24** with azides **27b**–**e** proceeded smoothly with the formation of triazole-substituted caffeine derivatives **28**–**32**. The reaction of alkyne **24** with tertiary azide **27f** required an additional selection of conditions for improving the conversion. The yield of the desired product **33** was increased to 69% by conducting the reaction at 90 °C in a sealed tube (Figure 2).

Our main focus was the introduction of a variety of nitrogen-substituted aminopropargyl functional moieties in the C-8 position of methylxanthines (Figure 1). The literature pathway for the preparation of those compounds follows the Pd-Cu-catalyzed cross-coupling reaction of nitrogen-substituted propargylamines with 8-iodo- [45] or 8-bromoxanthines [46,47]. However, none of the above methods has found broad application for the preparation of 8-(aminopropargyl)xanthine derivatives, possibly due to moderate yields and also the low availability of nitrogen-functionalized propargylamines. 

The copper(I) catalyzed one-pot three-component (A^3^-coupling) reaction among terminal alkyne, formaldehyde, and amine (the Mannich reaction) has become a popular approach to synthesizing propargylamines [48,49]. Therefore, the sequence of the Sonogashira cross-coupling reaction of 3-bromocaffeine **5** and the A^3^-coupling reaction of 8-ethynylcaffeine **24** has been exploited in our studies (Figure 3). The reaction of compound **24**, aq. formaldehyde (3 equiv.) and diethylamine **34** (3 equiv.) in THF in the presence of copper(I) iodide or copper(I) chloride (0.1 equiv.) at 75 °C (bath) proceeds smoothly, and after 3 h the alkyne was almost consumed and the desired 8-(3-(diethylamino)prop-1-yn-1-yl)-1,3,7-trimethyl-3,7-dihydro-1*H*-purine-2,6-dione **35** was formed. By using copper (II) acetate monohydrate (0.1 equiv.) as the catalyst in the above conditions, compound **35** was isolated in a yield of 94% after column chromatography (Figure 3). In this condition, 8-ethynylcaffeine **24** was reacted with secondary amines **36**–**38** to give the desired caffeine derivatives **39**–**41** in high isolated yields (57–96%). The reaction of pyrrolidine **42**, azepane **43**, azocane **44**, homomorpholine **45**, 4-methylpiperidine **46**, morpholine **47**, and *N*-substituted piperazines **48**–**50** with aq. formaldehyde and 8-ethynylcaffeine **24** in the presence of Cu(OAc)_2_ × H_2_O (conditions *a*) led to *N*-substituted 8-(1-(aminopropargyl))caffeine derivatives **51**–**59** in high yields (Figure 3). All the caffeine derivatives were purified by column chromatography (chloroform:ethanol, solvent mixture). So we found that this A^3^-coupling approach allows for the direct installation of nitrogen-functionalized aminopropargyl substituent at the C-8 position, providing efficient access to different 8-(1-aminopropargyl)trimethylxanthines. It is well known that the reported reaction is thought to proceed through the alkyne activation forming a copper acetylide. After a nucleophilic addition on the intermediate formed by the reaction of formaldehyde and a secondary amine, the propargylamine derivative is obtained [48]. We performed also a one-pot deprotection–A^3^-coupling tandem procedure for obtaining compound **35**. We found that the consecutive treatment of 8-(trimethylsilylethynyl)caffeine **26** in THF under argon flow with Bu_4_NF (1.1 equiv.), amine **34**, and aq. formaldehyde in the presence of Cu(OAc)_2_ × H_2_O as the catalyst (0.1 equiv.) gave the desired caffeine derivative **35** in the isolated yield 44%. No products of alkyne deprotection reaction (compound **24** or dimeric butadiyne) were observed in this condition. It should be noted that caffeine derivatives **51**, **52**, and **56** are known compounds, which were synthesized by the Sonogashira reaction of 8-bromocaffeine with subsequently substituted alkynes (yield 38–57%) [46,47]. Using an A^3^-coupling reaction between 8-ethynylcaffeine **24**, formaldehyde, and secondary amines seems to be more effective; the reaction proceeds with higher isolated yield by using more available and stable secondary amines as the key reagents. The stage of synthesis of ethynylamines necessary for a cross-coupling reaction can be problematic due to side reactions, as well as to purify. 

To further investigate the significance of the propynyl linker on enzyme inhibition assay, we synthesized caffeine derivatives **60**, **61** with N-methylpiperidinyl- or N-Boc-piperazinyl- substituent at the C-8 position (Figure 4). Compounds **60**, **61** (yield 78–88%), we prepared by the reaction of 8-bromocaffeine **5** with an excess (3 equiv.) of 4-methylpiperidine **46** or 1-Boc-piperazine **49** in DMF. 

Next, we concentrated our efforts on the design and synthesis of purine derivatives containing a nitrogen-substituted aminobut-2-ynyl residue in position N-1 or N-7 on the methylxanthine core. To our delight, we observed that the A^3^-coupling process was amenable to 1-(prop-2-yn-1-yl)-3,7-dimethylxanthine **62** obtained from the reaction of theobromine **2** with propargyl bromide **63** under basic conditions (Figure 5). The three-component reaction of **63** with aq. formaldehyde and secondary amines, including diisopropylamine **36**, azocane **44**, or 1-(2-(pyrrolidin-1-yl)ethyl)piperazine **50** in THF in the presence of copper(II) acetate monohydrate was very efficient, and occurred with the formation of nitrogen substituted 1-(4-(amino)but-2-yn-1-yl)-3,7-dimethyl- 3,7-dihydro-1*H*-purine-2,6-diones **64**–**66** (54–90%) (Figure 5).

A synthetic route was designed for the preparation of nitrogen-substituted 7-(aminobut-2-yn-1-yl)theophylline derivatives as depicted in Figure 6. The reaction of theophylline **3** with propargyl bromide **63** led to the formation of 7-(prop-2-yn-1-yl)-1,3-dimethyl-3,7-dihydro-1*H*-purine-2,6-dione **67**. The three-component reaction of the alkyne **67** with an excess of formaldehyde and appropriate secondary amines **36**, **44**, or **50** in the presence of a catalytic amount of Cu(OAc)_2_ × H_2_O in THF afforded the target compounds **68**–**70** in an excellent yield (Figure 6).

The composition and structure of all synthesized compounds were confirmed by IR, UV, ^1^H and ^13^C NMR spectroscopy, and mass spectrometry (HRMS). The purity of all compounds was checked additionally by thin-layer chromatography and elemental analysis. The ^1^H and ^13^C NMR spectra of all synthesized compounds agreed with their structures and contained one set of characteristic signals for the xanthine core and the corresponding substituent. The structures of 8-arylsubstituted and 8-triazolylsubstituted caffeine derivatives **18**, and **32** were established by X-ray structure analysis. The refined molecules are shown in Figure 3. Aryl substituent on C8 in structure **18** has an interplane angle with caffeine core equaling 72.1º and the bond length of C8-C1′ 1.479(2) Å being longer than the same one in **32**. The π…π–interactions of pyrimidine cycles form molecular dimers (Figure 3A, right), having the following geometric parameters: the intercentroid distances equaling 3.639(1) and distances from the centroid to plain 3.354(1) Å.

Notably, the triazole ring in structure **32** lays practically in the plain of caffeine (Figure 3B); the interplane angle between cycles is 1.3°, and the C8-C4 bond length equals 1.455 Å (the same as for a single bond in conjugated C=C-C=C [50]). Such an extended aromatic system and the presence of a hydroxy group lead to the formation of 1-D motifs along axes *c* (Figure 3B, right) due to interactions of the π-system, with the intercentroid distances equaling 3.497(2)–3.581(2), and distances from centroids to plain laying in the interval of 3.337(2)–3.366(2)^2x^ Å, together with hydrogen bond O3A-H…N2′ with parameters H…N 2.21, O…N 2.912(8) Å and O-H…N 144° [51].

### 2.2. Enzyme Inhibition

The synthesized 8-aryltrimethylxanthine derivatives **7**, **14**–**19**, 8-ethynylderivatives **24**, **26**, 8-triazolylcaffeines **28**–**33**, substituted at nitrogen atom 8-(3-(R-amino)prop-1-yn-1-yl)caffeine derivatives **35**, **39**–**41**, **51**–**59**, 8-(N-methylpiperidinyl)- **60**, 8-(N-Boc-piperazinyl)caffeine **61**, 1-(4-(amino)but-2-yn-1-yl)-3,7-dimethyl-3,7-dihydro- **64**–**66**, 7-(4-(amino)but-2-yn-1-yl)- 1,3-dimethyl-3,7-dihydro-1*H*-purine-2,6-diones **68**–**70**, and the starting caffeine **1** were evaluated for their ability to inhibit AChE from electric eel (EeAChE) using Ellman’s method [52]. The enzymatic activity is quantified indirectly by spectrophotometric determination of the 5-thio-2-nitrobenzoate concentration formed in the reaction between the reagent 5,5′-dithio-bis(2-nitrobenzoic acid) and thiocholine, a product of acetylthiocholine hydrolysis catalyzed by cholinesterases. The efficacy of inhibitors is expressed as IC_50_, i.e., the concentration required for 50% inhibition of the enzyme activity obtained from three independent experiments. Results are given in Table 1. Galantamine, a clinically used drug acting as a balanced inhibitor of AChE, was used as a standard for IC_50_ values comparison. The substituent at the C-8 position of caffeine has a great influence on AChE inhibition. IC_50_ values for AChE for 8-aryl-, 8-triazolyl-, and 8-alkynyltrimethylxanthines ranged from 21.3 to 531.0 μM (**31** and **28**, respectively). When compared to caffeine, only two compounds **22** and **31** produced equal IC_50_ values. Focused on 8-(3-aminoprop-1-yn-1-yl)caffeines, seven derivatives are more potent AChE inhibitors **35**, **39**, **52**, **53**, **55**, **58**, and **59**, and two are comparable **41**, **51** to caffeine (Table 1). Clearly, most of the derivatives **53**, **55**, **58**, and **59** are stronger inhibitors of AChE than galantamine. Characteristically, these four compounds have a cyclic (azocanyl, 4-methylpiperidinyl, 4-Boc-piperazinyl or 4-((2-(pyrrolidin-1-yl)ethyl)piperazinyl) substituent in the propargylamine moiety.

Drawing a comparison among N-substituted 8-(1-aminopropargyl)caffeines **35**, **39**–**41**, **51**–**59**, we can describe the following general SAR: For AChE inhibition, the presence of a cyclic substituent at the propargylamino moiety is essential. A remarkable increase in inhibition was observed for compound **53** with a bulk 8-(3-(azocan-1-yl)propargyl)substituent in the 8-th position of trimethylxanthine core (the best nanomolar AChE inhibitor, IC_50_ = 0.255 ± 0.016 µM). Oxygen substituents in the cycle (morpholino and homomorpholino, compounds **54**, **56**) decreased the activity. 8-(1-Aminopropargyl)caffeines with piperazinyl substituents **58**, **59** improved inhibition of AChE significantly (IC_50_ = 0.552 and 1.20 µM) when compared to galantamine (IC_50_ = 4.9 µM). Spacing the 1-Boc-piperazinyl substituent away from the xanthine core by a propargyl linker led to a great increase in the AChE inhibition (compared to compounds **58** (IC_50_ = 1.20 µM) and **61** (IC_50_ = 42.0 µM) (Table 1). 

The introduction of 1-(4-(aminobut-2-yn-1-yl) substituent in the theobromine core (compounds **64**–**66**) has improved the inhibition of AChE significantly. The 1-(4-(azocan-1-yl)but-2-yn-1-yl)-3,7-dimethyl-3,7-dihydro-1*H*-purine-2,6-dione (**65**) was the most active AChE inhibitor among all studied (IC_50_ = 0.089 µM); characteristically, all three 1-(4-(aminobut-2-yn-1-yl))dimethylxanthines **64**–**66** were superior in inhibition of AChE compared to galantamine. The high inhibition of **64**–**66** of AChE is of interest because the starting compound 1,3,7-trisubstituted xanthine derivative (DMPX) **62** was the first A_2A_ adenosine receptor antagonist reported in the literature having a *h*A_2A_ Ki = 4.1 µM [53]. The introduction of the 7-(4-(amino)but-2-yn-1-yl) in the theophylline core (compounds **68**–**70**) was also very useful for the construction of AChE inhibitors. Notably, in the series of these derivatives, compound **69** with the 7-(3-(azocan-1-yl)-but-2-yn-1-yl) substituent was more efficient than **68** and **70** derivatives in AChE inhibition; the bulk azocane substituent was helpful to the inhibitory activities (Table 1). 

The obtained data indicated that the AChE inhibitory activity of 8-aminopropynyl-, 1-, and 7-aminobutynylmethylxanthines is sensitive to the nature of the nitrogen functionalities in the substituent. To the best of our knowledge, this structural type of AChE inhibitor is presented herein for the first time, and our results confirmed the potential value of this class of biologically active substances. 

### 2.3. Molecular Docking Study

With the aim of obtaining some information about the possible interactions of the methylxanthine derivatives with the enzyme, which, in turn, could be useful in the design of more potent inhibitors, a molecular modeling study was performed comparing with donepezil, a nanomolar AChE inhibitor. The calculation of the interaction energy of donepezil in XRD model coordinates (score in place mode) gave an estimated binding energy of −14.806 kcal/mol. Molecular docking of the minimized donepezil molecule to the AChE active site showed a very close value of −14.817 kcal/mol (Table 2). Compounds **64**, **68**, **65**, **69**, and **55** show a theoretical affinity for the AChE active site at the level of donepezil. 

When comparing the conformations of donepezil in the XRD model and that obtained as a result of docking, the RMSD of the atomic coordinates was determined to be 0.363. Thus, the procedure of molecular docking under the conditions of the chosen AChE model makes it possible to quite accurately predict the position and conformation of the inhibitor. Molecular interactions of donepezil and active xanthine derivatives are shown in Figure 4. 

The active site of AChE is saturated with aromatic amino acids, which provided the main type of non-covalent interactions with inhibitors-stacking and interactions of the π-systems of Trp286, Tyr341, Phe338, Tyr337, and Trp86 amino acids with π-systems and individual atoms of inhibitors. In this regard, the aromatic systems of the benzyl and dihydroindene rings of donepezil are actively involved in the interactions (Figure 4A). Apparently, the hydrogen bond between the oxygen of the keto group associated with the dihydroindene ring and the amino acid residue Phe295 is important for the formation of a stable interaction between the inhibitor and the AChE active site. The xanthine core of the new derivatives is actively stacked with the Trp286 and Tyr341 π-systems (except for compound **66**). A common feature of all the most active compounds is the ability of the carbonyl groups of the purine cycle to form a hydrogen bond with Phe295. Due to the high saturation of xanthine with polar atoms, such a bond can arise at any orientation of the purine in the active site of AChE, depending on the site of the linker attachment. 

Analysis of the SAR of compounds differing in the position of attachment of the same type of substituent to the xanthine backbone of caffeine shows that, apparently, in the series of compounds with the diisopropylamino function: **64**, **68**, and **39**, the presence of a rotating bond in front of the propargylamine substituent has a significant effect on the theoretical affinity (Figure 4E). The most theoretically affine compounds in this series (**64** and **68**) are modified by adding an aminobut-2-inyl linker to diisopropylamino substituent at positions 1 or 7 of the xanthine core (Figure 4B,C), which allows the diisopropylaminobut-2-ynyl substituent to occupy a more advantageous position compared to compound **39** (Figure 4D), where the rigid aminopropargyl substituent is attached directly to the 8-th position of the core. The presence of a rigid acetylene bond in compound **39** causes the only possible position of the linear part, and (isopropylamino)propynyl substituent acquired the possibility of the formation of hydrophobic interactions of isopropyl moiety with Trp-86. When comparing compounds **65** and **69**, with a bulky azocane moiety at the but-2-ynyl substituent, it is more advantageous for binding to attach a substituent at position 1 of the xanthine backbone compared to position 7 (Figure 4F–H). The differences in the estimated binding energies may be due to the fact that different carbonyl groups form a hydrogen bond with Phe295: in the case of derivative **65**, this is the carbonyl oxygen in position 2 of the backbone, and for compound **69** – the carbonyl group in position C-6. The azocane substituent ensured the hydrophobic interaction with Trp86. 

For compound **66** (pyrrolidinoethylpiperazinyl)butynyl substituent at position 1 of the backbone), the most energetically favorable conformation occurs at the “reverse” position of the methylxanthine core compared to the typical position for other active derivatives (Figure 4I). With this orientation, no stacking occurs with amino acids characteristic of AChE inhibitors, and no hydrogen bond is formed with Phe295; however, the carbonyl group at C-2 forms a hydrogen bond with Glu202. In this case, it is worth considering other, less energetically favorable conformations in comparison with other active compounds. Hydrophobic interaction of substituent: piperazine ring with Phe-338 and pyrrolidine ring with Tyr341 and Trp286 are maintained (Figure 4I). Compound **59** ((pyrrolidinoethylpiperazinyl)prop-2-ynyl substituent at position 8 of the backbone) does not form a hydrogen bond with Phe295 (Figure 4J). The binding mode for the *h*AChE pocket is characterized by π-π interactions of the xanthine core with Tyr341. The internal ethylpiperazinyl substituent at the alkyne in the C-8 position formed hydrophobic interaction with Phe 338. The pyrrolidinoethylpiperazinyl substituent of this derivative is U-folded. Compared to donepezil, its structure is less flexible due to the presence of a rigid propargylamine linker (Figure 4K). However, the pyrrolidine ring at the end of the substituent is able to establish the interaction with Trp86 (Figure 4I).

This molecular modeling helped to distinguish the key interaction of substituted methylxanthines and AChE. Molecular dynamics was performed to investigate the stability of the protein-ligand complexes in time (during 100 nanoseconds) Appendix A).

**Donepezil**. During the simulation, the protein-ligand complex was stable, but interestingly, based on the RMSD-time relation, two segments of the simulation could be distinguished, with the first being from 5 to 38 ns and the second from 54 to 100 ns. It should be noted that it corresponds to the conformational change “chair to chair” of the piperidine fragment of the donepezil ligand (Appendix A).

**Compound 59**. In general, the protein-**59** complex was stable during the simulation; the insignificant deviations of ligand RMSD from 2 to 3 Å correspond to the conformational flexibility of the *N*-alkylpyrrolidine fragment of **59** (Appendix A).

**Compound 64**. Similar to **59**, the protein-**64** complex was stable during the simulation as well. As could be seen from the RMSD-time relation, a ligand reversible conformation change occurred. Analyzing the trajectory of the protein–ligand complex, we concluded that it was a result of conformational interconversion of the *N*-diisopropyl fragment of **64**. In particular, a rotation took place across the -N-C- bond with a corresponding conformational change of the *N*-diisopropyl fragment from anti-gauche to gauche-anti (Appendix A).

**Compound 69**. As could be seen, the protein-**69** complex was stable for the first 35 nanoseconds of the simulation, and after that compound **69** partially left the binding pocket. During the period from 35 to 55 ns, the ligand was bound with protein by an azocyl fragment. It seemed to be a formation of π-cation interaction between the protonated tertiary amino group and the side chain of Trp286. On the other hand, the formation of hydrogen bonds between the Tyr77 hydroxyl group and the C6-carbonyl group of the purine core is supposed to be possible. During the following simulation, after 55 ns, the dissociation of the protein–ligand complex occurred. Interestingly, after 90 ns, the ligand binds an opposite side of the protein in the region of 61–64 amino acids with the following dissociation after 98 nanoseconds (Appendix A).

## 3. Materials and Methods

### 3.1. Chemistry

#### General Information

^1^H and ^13^C NMR spectra were acquired on ‘Bruker AV 300′ 300.13 (^1^H) and 75.47 MHz (^13^C), respectively, (compounds **26**, **30**, **31**, **32**, **35**, **40**, **59**), ‘Bruker AV 400′ 400.13 (^1^H) and 100.61 MHz (^13^C) (**15**, **18**, **19**, **22**, **24**, **33**, **39**, **41**, **52**, **58**), ‘Bruker DRX-500′ 500.13 (^1^H) and 125.76 MHz (^13^C), (**16**, **23**, **55**, **60**, **61**) instruments. The ^1^H spectra for compounds **28**, **29**, **53**, **54**, **56**, and **57** were obtained on ‘Bruker AV 400′, and ^13^C spectra–on ‘Bruker DRX-500′. Deuterochloroform (CDCl_3_) was used as a solvent, with residual CHCl_3_ (δ_H_ = 7.24 ppm) or CDCl_3_ (δ_C_ = 76.9 ppm) being employed as internal standards. NMR signal assignments were carried out with the aid of a combination of 1D and 2D NMR techniques that included ^1^H, ^13^C, COSY, HSQC, and HMBC spectra. IR absorption spectra were recorded on a Vector 22 FT-IR spectrometer in KBr pellets. The UV spectra were obtained on an HP 8453 UV Vis spectrometer (Hewlett-Packard, Waldbronn, Germany). Melting points were determined using the Mettler Toledo FP900 (USA) thermosystem. HRMS spectra were recorded on a Thermo Scientific DFS mass spectrometer (evaporator temperature 190–250 °C, EI ionization at 70 eV). Elemental analysis was carried out on an 1106 Elemental analysis instrument (Carlo-Erba, Milan, Italy). The X-ray diffraction experiments for crystals **18** and **32** were carried out at ambient conditions on a Bruker KAPPA APEX II diffractometer (graphite-monochromated Mo Kα radiation). Reflection intensities were corrected for absorption by the *SADABS* program [54]. The structures were solved by direct methods using *SHELXS-97* [55] for **18** and the *SHELXT* 2014/5 [56] program for **32** and refined by anisotropic (isotropic for all H atoms and O in minor part of disordered OH group of **32**) full-matrix least-squares method against *F*^2^ of all reflections by *SHELXL2018*/3 [57]. The positions of the hydrogen were calculated geometrically and refined in a riding model. The OH group of compound **1** is disordered due to rotation, approximately as 9:1.

The reaction progress and the purity of the obtained compounds were monitored by TLC on Silufol UV-254 plates (CHCl_3_-EtOH, 9:1; detection under UV light or by treatment with iodine vapor). Products were isolated by column chromatography on silica gel 60 (0.063–0.200 mm, Merck KGaA, Darmstadt, Germany) eluting with indicated solvent systems. The chemicals used: caffeine **1**, theobromine **2**, theophylline **3**, arylboronic acids **6**, **8**–**13**, furan-3-boronic acid **20**, indole-5-boronic acid **21**, (trimethylsilyl)acetylene **25**, azides **27a-f**, formalin (30% formaldehyde in aq. solution), diethylamine **34**, diisopropylamine **36**, dibutylamine **37**, dicyclohexylamine **38**, pyrrolidine **42**, azepane **43**, azocane **44**, homomorpholine **45**, 4-methylpiperidine **46**, morpholine **47**, *N*-methylpiperazine **48**, *N*-Boc-piperazine **49**, 1-(2-(pyrrolidin-1-yl)ethyl)piperazine **50**, propargyl bromide **63**, CuI, CuCl, Cu(OAc)_2_ × H_2_O, tetra-*n*-butylammonium fluoride trihydrate (TBAF × 3H_2_O), Bu_4_NBr, Pd(PPh_3_)_4_, DIPEA, 5,5′-dithio-bis-(2-nitrobenzoic acid) (DTNB, Ellman’s reagent), were purchased from Aldrich (St. Louis, MO, USA) or Alfa Aesar (GmbH, Karlsruhe, Germany). 5-Chlorocaffeine **4** and 5-bromocaffeine **5** were obtained according to published procedures [39,40]. Solvents (DMF, THF, toluene, MeCN, CHCl_3_, CH_2_Cl_2_) and Et_3_N were purified by standard methods. Copies of NMR spectra (^1^H and ^13^C) are given in Appendix A. 

### 3.2. Synthesis and Spectral Data

#### 3.2.1. General Method for the Preparation of 8-Aryl(hetaryl)-1,3,7-trimethyl- 3,7-dihydro-1H-purine-2,6-diones (**7**, **14**–**19**, **22**, **23**)

8-Chlorocaffeine **4** (115 mg, 0.5 mmol) was dissolved in toluene (2 mL) in a microwave vial under argon. Boronic acid (1.2 equiv.) **6**, **8**–**13**, **20** or **21**, (Pd(PPh_3_)_4_ (1–5 mol %), TBAB (161 mg, 0.5 mmol), and K_2_CO_3_ (0.21 g, 1.5 mmol) in H_2_O (2 mL) were successively added. The mixture was stirred for 1.5 h in a microwave reactor at 130 °C. After cooling, the reaction mixture was diluted with NaCl (10 mL), the organic layer was separated, and the aqueous layer was extracted with EtOAc (3 × 20 mL). The combined organic layers were washed with H_2_O (2 × 20 mL), dried over MgSO_4_, and evaporated. The residue was purified by column chromatography (eluent petroleum ether-Et_2_O, 2:1) to give compounds **7**, **14**–**19**, **22**, **23**.

##### 1,3,7-Trimethyl-8-phenyl-3,7-dihydro-1*H*-purine-2,6-dione (**7**)

Yield 70%. White crystals, M.p. 180–181 °C (EtOH); M.p. 178 °C [41]. ^1^H NMR, ^13^C NMR, and MS data are given in [41]. 

##### 1,3,7-Trimethyl-8-*o*-tolyl-3,7-dihydro-1*H*-purine-2,6-dione (**14**)

Yield 87%. White solid, M.p. 194.5 °C (EtOH, decomp.); M.p. 193–194 °C [58]. ^1^H NMR, ^13^C NMR, and MS data are given in [58]. 

##### 8-(2-Aminophenyl)-1,3,7-trimethyl-3,7-dihydro-1*H*-purine-2,6-dione (**15**)

Yield 53%. White solid. M.p. 195.6 °C (EtOH, decomp.). ^1^H NMR (400 MHz, CDCl_3_, δ, ppm): 3.40 (3H, s, NCH_3_), 3.58 (3H, s, NCH_3_), 3.97 (3H, s, NCH_3_), 4.48 (2H, s, NH_2_), 6.84 (1H, d, J = 8.2 Hz, H-3′), 7.26–6.92 (2H, m, H-4′,5′), 7.45 (1H, d, J = 7.8 Hz, H-6′); ^13^C NMR (101 MHz, CDCl_3_, δ, ppm): 27.9 (NCH_3_), 29.7 (NCH_3_), 34.0 (N CH_3_), 107.7 (C-5), 111.4 (C-1′), 116.8 (C-3′), 117.5 (C-5′), 129.8 (C-6′), 131.4 (C-4′), 146.7 (C-2′), 147.7 (C-4), 150.7 (C-8), 151.5 (C-2), 155.2 (C-6); IR (KBr, ν, cm^−1^): 3410 (NH_2_), 1655, 1693 (C=O), 1626, 1543, 1508, 1475, 746 (C=C,C=N); UV (MeCN) λ_max_ (lgε): 225 (4.49), 291 (4.04), 318 (3.91) nm. Anal. calcd for C_14_H_15_N_5_O_2_, %: C, 58.94; H, 5.30; N, 24.55; found, %: C, 58.57; H, 5.56; N, 24.22.

##### 8-(3-Methoxyphenyl)-1,3,7-trimethyl-3,7-dihydro-1*H*-purine-2,6-dione (**16**)

Yield 60%. White crystals. M.p. 171.0–171.6 °C (EtOH). ^1^H NMR (500 MHz, CDCl_3_, δ, ppm): 3.40 (3H, s, NCH_3_), 3.60 (3H, s, NCH_3_), 3.85 (3H, s, OCH_3_), 4.03 (3H, s, NCH_3_), 6.82 (1H, dd, J = 8.2, 1.8 Hz, H-4′), 7.28 (1H, dd, 8.2, 7.8 Hz, H-5′), 7.31 (1H, d, J = 1.8 Hz, H-2′), 7.60 (1H, d, J = 7.8 Hz, H-6′); ^13^C NMR (126 MHz, CDCl_3_, δ, ppm): 27.9 (NCH_3_), 29.7 (NCH_3_), 33.8 (NCH_3_), 55.3 (OCH_3_), 108.4 (C-5), 114.6 (C-2′), 116.0 (C-4′), 121.2 (C-6′), 129.3 (C-5′), 129.9 (C-1′), 148.0 (C-4), 151.6 (C-2), 151.8 (C-8), 155.4 (C-6), 159.7 (C-3′); IR (KBr, ν, cm^−1^): 1659, 1691 (C=O), 1605, 1500, 1543, 1471, 1441, 747 (C=C, C=N); UV (MeCN) λ_max._ (lgε): 217 (4.56), 298 (4.23) nm. Anal. calcd for C_15_H_16_N_4_O_3_, %: C, 59.99; H, 5.37; N, 18.66; found, %: C, 59.85; H, 5.84; N, 18.97.

##### 8-(4-Methoxyphenyl)-1,3,7-trimethyl-3,7-dihydro-1*H*-purine-2,6-dione (**17**)

Yield 53%. White crystals, M.p. 168.6 °C (EtOH, decomp.); M.p. 168–169 °C [58]. ^1^H NMR, ^13^C NMR, and MS data are given in [58].

##### 8-(2,3-Dimethoxyphenyl)-1,3,7-trimethyl-3,7-dihydro-1*H*-purine-2,6-dione (**18**)

Yield 27%. White crystals. M.p. 200.7 °C (EtOH). ^1^H NMR (400 MHz, CDCl_3_, δ, ppm): 3.41 (3H, s, NCH_3_), 3.60 (3H, s, NCH_3_), 3.65 (3H, s, OCH_3_), 3.81 (3H, s, OCH_3_), 3.90 (3H, s, NCH_3_), 7.02 (1H, dd, J = 8.0, 1.8 Hz, H-4′), 7.06 (1H, dd, J = 7.6, 1.8 Hz, H-6′), 7.17 (1H, dd, J = 8.0, 7.6 Hz, H-5′); ^13^C NMR (101 MHz, CDCl_3_, δ, ppm): 27.9 (NCH_3_), 29.7 (NCH_3_), 33.3 (NCH_3_), 55.9 (OCH_3_), 61.3 (OCH_3_), 108.2 (C-5), 114.5 (C-4′), 122.8 (C-1′), 122.9 (C-5′), 124.7 (C-6′), 147.4, 148.0 (C-2′,3′), 149.8 (C-4), 151.6 (C-2), 152.7 (C-8), 155.4 (C-6); IR (KBr, ν, cm^−1^): 1657, 1707 (C=O), 1047 (C-O-C), 1543, 1481, 748 (C=C, C=N); UV (MeCN) λ_max_ (lgε): 217 (4.53), 286 (4.20) nm. Anal. calcd for C_16_H_18_N_4_O_4_, %: C, 58.17; H, 5.49; N, 16.96; found, %: C, 58.23; H, 5.99; N, 17.05. Crystallographic data for compound (18): C_16_H_18_N_4_O_4_, M 330.34, monoclinic, P2_1_/n, a 8.0807(4), b 24.784(1), c 8.3115(3) Å, β 107.255(2), V 1589.6(1) Å^3^, Z 4, D_calcd_ 1.380 g·cm^–3^, μ(Mo-Kα) 0.102 mm^–1^, F(000) 696, (θ 2.70–27.12°, completeness 99.6%), T 296(2) K, colorless, (0.96 × 0.50 × 0.30) mm^3^, transmission 0.6957–0.7455, 21885 measured reflections in index range −9 <= h <= 10, −31 <= k <= 31, −10 <= l <= 10, 3511 independent (R_int_ 0.0 305), 222 parameters, R_1_ 0.0582 (for 2983 observed I > 2σ(I)), wR_2_ = 0. 1628 (all data), GOOF 1.091, largest diff. peak and hole 0.419 and −0.276 e.A^−3^. Crystallographic data for compound (18) has been deposited at the Cambridge Crystallographic Data Centre as supplementary publication no. 2208208. Copy of the data can be obtained free of charge, on application to CCDC, 12 Union Road, Cambridge CB21EZ, UK (fax: +44 122 3336033 or e-mail: deposit@ccdc.cam.ac.uk; internet: www.ccdc.cam.ac.uk).

##### 8-(3,4,5-Trimethoxyphenyl)-1,3,7-trimethyl-3,7-dihydro-1*H*-purine-2,6-dione (**19**)

Yield 47%. White crystals. M.p. 214.1^o^C (EtOH, decomp.). ^1^H NMR (400 MHz, CDCl_3_, δ, ppm): 3.40 (3H, s, NCH_3_), 3.60 (3H, s, NCH_3_), 4.03 (3H, s, NCH_3_), 3.90 (6H, s, OCH_3_), 3.88 (3H, s, OCH_3_), 6.83 (2H, s, H-2′,6′); ^13^C NMR (101 MHz, CDCl_3_, δ, ppm): 27.9 (NCH_3_), 29.7 (NCH_3_), 33.8 (NCH_3_), 56.2 (2 × OCH_3_), 60.9 (OCH_3_), 106.5 (C-2′,6′), 108.3 (C-5), 123.4 (C-1′), 140.0 (C-4′), 148.0 (C-4), 151.6 (C-2), 151.9 (C-3′,5′), 153.4 (C-6), 155.4 (C-8). IR (KBr, ν, cm^−1^): 1664, 1707 (C=O), 1587, 1541, 1485, 1470, 1454, 695 (C=C, C=N); UV (MeCN) λ_max._ (lgε): 222 (4.57), 251 (4.00), 303 (4.29) nm. Anal. calcd for C_17_H_20_N_4_O_5_, %: C, 56.66; H, 5.59; N, 15.55; found, %: C, 57.05; H, 5.37; N, 15.36.

##### 8-(Furan-3-yl)-1,3,7-trimethyl-3,7-dihydro-1*H*-purine-2,6-dione (**22**)

Yield 33%. Yellowish solid. M.p. 205.7 °C (EtOH, decomp.). ^1^H NMR (400 MHz, CDCl_3_, δ, ppm): 3.37 (3H, s, NCH_3_), 3.56 (3H, s, NCH_3_), 4.05 (3H, s, NCH_3_), 6.85 (1H, s, H-4′), 7.54 (1H, s, H-5′), 7.92 (1H, s, H-2′); ^13^C NMR (101 MHz, CDCl_3_, δ, ppm): 27.8 (NCH_3_), 29.6 (NCH_3_), 33.0 (NCH_3_), 107.9 (C-5), 109.7 (C-4′), 132.5 (C-3′), 143.9 (C-2′), 145.8 (C-5′), 148.1 (C-4), 151.5 (C-2), 154.3 (C-8), 155.2 (C-6); IR (KBr, ν, cm^−1^): 1695, 1655 (C=O), 1567, 1539, 1464, 1429, 746 (C=C, C=N); UV (MeCN) λ_max._ (lgε): 213 (4.41), 298 (4.16) nm. Anal. calcd for C_12_H_12_N_4_O_3_, %: C, 55.38; H, 4.65; N, 21.53; found, %: C, 55.48; H, 5.03; N, 21.24.

##### 8-(1*H*-Indol-5-yl)-1,3,7-trimethyl-3,7-dihydro-1*H*-purine-2,6-dione (**23**)

Yield 52%. White solid. M.p. 292.7 °C (EtOH, decomp.). ^1^H NMR (500 MHz, CDCl_3_, δ, ppm): 3.33 (3H, s, NCH_3_), 3.53 (3H, s, NCH_3_), 3.95 (3H, s, NCH_3_), 6.51 (1H, d, J = 3.0 Hz, H-3′), 7.22 (1H, d, J = 3.0 Hz, H-2′), 7.44 (1H, d, J = 8.5 Hz, H-7′), 7.63 (1H, s, H-4′), 7.81 (1H, d, J = 8.5 Hz, H-6′); ^13^C NMR (126 MHz, CDCl_3_, δ, ppm): 27.9 (NCH_3_), 29.7 (NCH_3_), 33.9 (NCH_3_), 102.5 (C-3′), 108.2 (C-5), 111.7 (C-7′), 118.7 (C-6′), 122.0 (C-4′), 124.4 (C-2′), 126.1 (C-3a′), 127.7 (C-5′), 136.8 (C-7a′), 148.2 (C-4), 151.8 (C-2), 154.4 (C-8), 155.5 (C-6); IR (KBr, ν, cm^−1^): 1448, 1543, 1620, 1647, 731 (C=N, C=C), 1689 (C=O), 3273 (NH); UV (MeCN) λ_max._ (lgε): 240 (4.56), 278 (4.25), 306 (4.29) nm. Anal. calcd for C_16_H_15_N_5_O_2_, %: C, 62.13; H, 4.89; N, 22.64; found, %: C, 61.32; H, 4.81; N, 22.36.

#### 3.2.2. 1,3,7-Trimethyl-8-((trimethylsilyl)ethynyl)-3,7-dihydro-1*H*-purine-2,6-dione {8-[(trimethylsilyl)ethynyl]-caffeine} (**26**)

A mixture of 8-bromocaffeine **5** (500 mg, 1.8 mmol), Pd(PPh_3_)_4_ (10 mg, 0.009 mmol), CuI (1.7 mg, 0.009 mmol), (trimethylsilyl)acetylene **25** (265 mg, 2.7 mmol), and Et_3_N (0.27 g, 2.7 mmol) in toluene (15 mL) was stirred at 95 °C (bath) for 6 h under a stream of argon. After cooling, the solvent was removed under reduced pressure, and the residue was treated with water (10 mL) and extracted with CHCl_3_ (3 × 20 mL). The combined organic solution was dried over MgSO_4_, filtered, and evaporated in vacuo. The residue was purified by column chromatography (eluent CHCl_3_-EtOH, 100:1→95:1) to afford compound **26** (495 mg, yield 93%). White powder. M.p. 166.1 °C (decomp.). ^1^H NMR (300 MHz, CDCl_3_, δ, ppm): 0.27 (9H, s, Si(CH_3_)_3_), 3.37 (3H, s, NCH_3_), 3.53 (3H, s, NCH_3_), 3.98 (3H, s, NCH_3_); ^13^C NMR (126 MHz, CDCl_3_,δ, ppm): −0.82 (Si(CH_3_)_3_), 27.79 (NCH_3_), 29.55 (NCH_3_), 32.94 (NCH_3_), 90.90 (C-11), 104.85 (C-10), 107.46 (C-5), 135.02 (C-8), 147.26 (C-4), 151.26 (C-2), 154.62 (C-6); IR (KBr, ν, cm^–1^): 2164 (C≡C), 1711, 1664 (C=O), 1601, 1545, 1477, 760, 744 (C=C, C=N); UV (EtOH) λ_max_ (lgε): 234 (4.44), 307 (4.29) nm. HR-MS, *m*/*z* (I*_rel._*, %): 290 (100), 276 (14), 275 (80), 143 (14), 142 (10), 82 (13), 67 (23), 58 (21), 57 (11), 56 (9); Calcd for C_13_H_18_N_4_O_2_Si: 290.1194; found [M]^+^
*m*/*z*: 290.1200. Anal. calcd for C_13_H_18_N_4_O_2_Si, %: C, 53.77; H, 6.25; N, 19.29; found, %: C, 53.87; H, 6.15; N, 19.40.

#### 3.2.3. 8-Ethynyl-1,3,7-trimethyl-3,7-dihydro-1*H*-purine-2,6-dione {8-Ethynylcaffeine} (**24**)

A stirred solution of compound **26** (3.00 g, 10.3 mmol) in dichloromethane (25 mL), was treated with a solution of tetra-*n*-butylammonium fluoride (12.4 mL of 0.1 M solution in THF, 12.3 mmol), and the mixture was stirred at 20 °C for 2 h. After completion based on TLC, the solvent was removed under reduced pressure, and the residue was treated with water (10 mL) and extracted with CHCl_3_ (3 × 30 mL). The combined organic solution was dried over MgSO_4_, filtered, and evaporated in vacuo. After column chromatography (eluent CHCl_3_-EtOH, 100:0→50:1) compound **24** (1.99 g, yield 88%) was isolated as a cream powder. 212 °C (decomp.). M.p. 214–215 °C [44]. ^1^H NMR (400 MHz, CDCl_3_, δ, ppm): 3.36 (s, 3H, NCH_3_), 3.52 (1H, s, H-11), 3.56 (3H, s, NCH_3_), 4.00 (3H, s, NCH_3_). ^13^C NMR (101 MHz, CDCl_3_, δ, ppm): 27.89 (NCH_3_), 29.62 (NCH_3_), 33.03 (NCH_3_), 71.37 (C-11), 85.32 (C-10), 107.80 (C-5), 134.27 (C-8), 147.31 (C-4), 151.33 (C-2), 154.70 (C-6); IR (KBr, ν, cm^–1^): 3219, 2955, 2118 (C≡C), 1707, 1657 (C=O), 1599, 1547, 1473, 1419, 758, 744 (C=C, C=N); UV (EtOH), λ_max_, (lgε): 226 (4.37), 299 (4.11) nm. HR-MS, *m/z* (I*_rel_*, %): 218 (100), 181 (22), 133 (23), 131 (42), 100 (14), 82 (30), 69 (62), 67 (76), 56 (15), 52 (16); Calcd for C_10_H_10_N_4_O_2_: 218.0798; found [M]^+^
*m*/*z:* 218.0804. Anal. calcd for C_10_H_10_N_4_O_2_, %: C, 55.04; H, 4.62; N, 25.68; found, %: C, 54.74; H, 4.48; N, 25.36.

#### 3.2.4. General Method for the Preparation of 8-(1,2,3-Triazol-4-yl)-3,7-дигидрo-1*H*-purine-2,6-diones (**28**–**33**)

A solution of 8-ethynylcaffeine **24** (300 mg, 1.4 mmol) in acetonitrile (10 mL) was treated with CuI (26 mg, 10 mol%), DIPEA (0.837 mL, 3.5 equiv.) and corresponding azide **27a**–**e** (1 equiv.). The reaction mixture was stirred at 50 °C in argon for 24 h. After cooling the solvent was evaporated, and the residue was treated with brine (10 mL) and extracted with CHCl_3_ (3 × 15 mL). The combined organic solution was dried over MgSO_4_, filtered, and evaporated in vacuo. After column chromatography (eluent CHCl_3_-EtOH, 100:0→20:1) compounds **28**–**32** were isolated. Compound **33** was obtained by heating the reaction mixture of **24**, CuI, DIPEA, and azide **27f** in a sealed tube to 90 °C for 24 h. 

##### 1,3,7-Trimethyl-8-(1-(*p*-tolyl)-1*H*-1,2,3-triazol-4-yl)-3,7-dihydro-1*H*-purine-2,6-dione (**28**)

Yield 84%. Yellow powder. M.p. 265.4 °C (decomp.). ^1^H NMR (400 MHz, CDCl_3_, δ, ppm): 2.41 (3H, s, CH_3–_4″), 3.38 (3H, s, NCH_3_), 3.56 (3H, s, NCH_3_), 4.45 (3H, s, NCH_3_), 7.33 (2H, d, J = 8.4 Hz, H-3″,5″), 7.65 (2H, d, J = 8.4 Hz, H-2″,6″), 8.57 (1H, s, H-5′); ^13^C NMR (126 MHz, CDCl_3_, δ, ppm): 21.01 (CH_3_-C4″), 27.86 (NCH_3_), 29.57 (NCH_3_), 34.30 (NCH_3_), 108.30 (C-5), 120.29 (C-3″,5″), 122.79 (C-5′), 130.29 (C-2″,6″), 133.82 (C-1″), 139.60 (C-4″), 139.62 (C-8), 142.46 (C-4′), 148.00 (C-4), 151.47 (C-2), 155.13 (C-6); IR (KBr, ν, cm^–1^): 3444, 3165, 3082, 2955, 1701 (C=O), 1655 (C=O), 1547, 1516, 1444, 822, 748 (C=C, C=N); UV (EtOH) λ_max_, (lgε): 223 (4.45), 304 (4.38) nm. HR-MS, *m*/*z* (I*_rel._*, %): 351 (71), 331 (20), 324 (22), 323 (100), 322 (44), 308 (32), 232 (20), 91 (17), 82 (29), 67 (39); Calcd for C_17_H_17_N_7_O_2_: 351.1438; found [M]^+^
*m*/*z:* 351.1441. Anal. calcd for C_17_H_17_N_7_O_2_, %: C, 58.11; H, 4.88; N, 27.90; found, %: C, 57.70; H, 4.76; N 27.69.

##### 1,3,7-Trimethyl-8-(1-(4-nitrophenyl)-1*H*-1,2,3-triazol-4-yl)-3,7-dihydro-1*H*-purine- 2,6-dione (**29**)

Yield 70%. Orange powder. M.p. 290.1–291.4 °C (decomp.). ^1^H NMR (400 MHz, CDCl_3_, δ, ppm): 3.42 (3H, s, NCH_3_), 3.59 (3H, s, NCH_3_), 4.50 (3H, s, NCH_3_), 8.06 (2H, d, J = 9.0 Hz, H-3″,5″), 8.46 (2H, d, J = 9.0 Hz, H-2″,6″), 8.72 (1H, s, H-5′); ^13^C NMR (126 MHz, CDCl_3_, δ, ppm): 27.94 (NCH_3_), 29.60 (NCH_3_), 34.38 (NCH_3_), 108.59 (C-5), 120.65 (C-2″,6″), 122.78 (C-5′), 125.60 (C-3″,5″), 140.27 (C-8), 140.55 (C-1″), 141.67 (C-4′), 147.52 (C-4), 147.98 (C-4″), 151.46 (C-2), 155.18 (C-6); IR (KBr, ν, cm^–1^1705, 1666 (C=O), 1595, 1344, 854 (NO_2_), 1545, 1525, 1448, 748 (C=C, C=N); UV (EtOH) λ_max_, (lgε): 222 (4.23), 305 (4.18) nm. HR-MS, m/z (I_rel._, %): 382 (100), 354 (45), 308 (44), 307 (19), 85 (28), 83 (44), 82 (42), 67 (69), 47 (16), 42 (22); Calcd for C_16_H_14_N_8_O_4_, %: 382.1133; found [M]^+^
*m*/*z*: 382.1132. Anal. calcd for C_16_H_14_N_8_O_4_: C, 50.26; H, 3.69; N, 29.31; found, %: C, 50.60; H, 3.45; N, 29.01.

##### 8-(1-Benzyl-1*H*-1,2,3-triazol-4-yl)-1,3,7-trimethyl-3,7-dihydro-1*H*-purine-2,6-dione (**30**)

Yield 65%. White powder. M.p. 243.1–248.2 °C. ^1^H NMR (300 MHz, CDCl_3_, δ, ppm): 3.37 (3H, s, NCH_3_), 3.51 (3H, s, NCH_3_), 4.41 (3H, s, NCH_3_), 5.58 (2H, s, H-6′), 7.29–7.40 (5H, m, Ph), 8.06 (1H, s, H-5′); ^13^C NMR (75 MHz, CDCl_3_, δ, ppm): 27.87 (NCH_3_), 29.56 (NCH_3_), 34.28 (NCH_3_), 54.43 (C-6′), 108.22 (C-5), 124.76 (C-5′), 129.07 (C-4″), 129.22 (C-2″,6″), 128.25 (C-3″,5″), 133.32 (C-1″), 139.34 (C-8), 142.70 (C-4′), 148.04 (C-4), 151.52 (C-2),155.20 (C-6); IR (KBr, ν, cm^–1^): 1705, 1666 (C=O), 1545, 1441 (C=C, C=N), 744 (C=C); UV (EtOH) λ_max_, (lgε): 231 (4.38), 300 (4.34) nm. HR-MS, *m*/*z* (I*_rel._*, %): 351 (41), 322 (59), 91 (100), 69 (53), 67 (61), 57 (58), 55 (55), 43 (82), 41 (69), 18 (54); Calcd for C_17_H_17_N_7_O_2_: 351.1438; found [M]^+^
*m*/*z:* 351.1439. Anal. calcd for C_17_H_17_N_7_O_2_, %: C, 58.11; H 4.88; N 27.90; found, %: C, 57.95; H, 4.88; N, 27.75.

##### 8-(1-Butyl-1*H*-1,2,3-triazol-4-yl)-1,3,7-trimethyl-3,7-dihydro-1*H*-purine-2,6-dione (**31**)

Yield 74%. White powder. M.p. 240.7–241.9 °C. ^1^H NMR (300 MHz, CDCl_3_, δ, ppm): 0.95 (3H, t, J = 7.2 Hz, H-9′), 1.37 (2H, sext, J = 7.2 Hz, H-8′), 1.93 (2H, sext, J = 7.2 Hz, H-7′), 3.38 (3H, s, NCH_3_), 3.54 (3H, s, NCH_3_), 4.41 (3H, s, NCH_3_), 4.42 (2H, t, J = 7.1 Hz, H-6′), 8.14 (1H, s, H-5′); ^13^C NMR (126 MHz, CDCl_3_, δ, ppm): 13.26 (C-9′), 19.48 (C-8′), 27.81 (NCH_3_), 29.53 (NCH_3_), 31.93 (C-7′), 34.21 (NCH_3_), 50.22 (C-6′), 108.11 (C-5), 124.67 (C-5′), 139.17 (C-8), 142.81 (C-4′), 147.98 (C-4), 151.46 (C-2), 155.12 (C-6). IR (KBr, ν, cm^–1^): 3444, 3144, 2958, 2874, 1707 (C=O), 1666 (C=O), 1547, 1443, 1288, 1047, 995, 744. UV (EtOH) λ_max_, (lgε): 230 (4.28), 300 (4.24) nm. HR-MS, *m*/*z* (I_rel._, %): 317 (100), 252 (21), 247 (23), 246 (31), 112 (26), 69 (21), 67 (31), 55 (23), 43 (21), 41 (35); Calcd for C_14_H_19_N_7_O_2_: 317.1595; found [M]^+^
*m*/*z*: 317.1601. Anal. calcd for C_14_H_19_N_7_O_2_, %: C, 52.99; H, 6.03; N, 30.90; found, %: C, 52.68; H, 5.77; N, 30.59.

##### 8-(1-(2-Hydroxyethyl)-1*H*-1,2,3-triazol-4-yl)-1,3,7-trimethyl-3,7-dihydro-1*H*-puri- ne-2,6-dione (**32**)

Yield 38%. White crystals. M.p. 232.0 °C (decomp.). ^1^H NMR (300 MHz, CDCl_3_, δ, ppm): 2.63 (1H, br.s, OH), 3.35 (3H, s, NCH_3_), 3.52 (3H, s, NCH_3_), 3.99 (2H, t, J = 5.0 Hz, C-7′), 4.36 (3H, s, NCH_3_), 4.53 (2H, t, J = 5.0 Hz, C-6′), 8.34 (1H, s, H-5′); ^13^C NMR (126 MHz, CDCl_3_, δ, ppm): 155.11 (C-6), 151.49 (C-2), 147.85 (C-4), 142.93 (C-4′), 138.50 (C-8), 126.25 (C-5′), 108.17 (C-5), 60.38 (C-7′), 52.81 (C-6′), 34.18 (NCH_3_), 29.54 (NCH_3_), 27.87 (NCH_3_); IR (KBr, ν, cm^–1^): 3433 (OH), 3142, 2955, 1705, 1668 (C=O), 1063, 1039 (C-O-C), 1545, 1443, 744 (C=N, C=C); UV (EtOH) λ_max_, (lgε): 230 (4.29), 299 (4.23) nm. HR-MS, *m*/*z* (I*_rel._*, %): 305 (100), 246 (27), 194 (52), 82 (22), 67 (57), 45 (39), 43 (22), 42 (24), 31 (43), 18 (25); Calcd for C_12_H_15_N_7_O_3_: 305.1231; found [M]^+^
*m*/*z:* 305.1234. Anal. calcd for C_12_H_15_N_7_O_3_, %: C, 47.21; H, 4.95; N, 32.12; found, %: C, 47.66; H, 5.21; N 32.54. Crystallographic data for compound (**32**): C_12_H_15_N_7_O_3_, *M* 305.31, orthorhombic, *Pccn*, *a* 14.420(2), *b* 27.125(4), *c* 6.9712(9) Å, *V* 2726.7(6) Å^3^, *Z* 8, *D*_calcd_ 1.487 g·cm^–3^, *μ*(Mo-*K*α) 0.112 mm^–1^, F(000) 1280, (θ 2.66–25.13°, completeness 98.9%), T 296(2) K, light-brown, (1.00 × 0.20 × 0.01) mm^3^, transmission 0.7507–0.9281, 31114 measured reflections in index range −17 <= h <= 17, −32 <= k <= 32, −5 <= l <= 8, 2416 independent (*R*_int_ 0.0432), 205 parameters, *R*_1_ 0.0721 (for 1686 observed *I >* 2*σ*(*I*)), *wR*_2_ = 0.211 (all data), GOOF 1.087, largest diff. peak and hole 0.507 and −0.482 e.A^−3^. Crystallographic data for compound **32** has been deposited at the Cambridge Crystallographic Data Centre as supplementary publication no. CCDC 2179477. Copy of the data can be obtained free of charge, on application to CCDC, 12 Union Road, Cambridge CB21EZ, UK (fax: +44 122 3336033 or e-mail: deposit@ccdc.cam.ac.uk; internet: www.ccdc.cam.ac.uk).

##### 8-(1-(*tert*-Butyl)-1*H*-1,2,3-triazol-4-yl)-1,3,7-trimethyl-3,7-dihydro-1*H*-purine-2,6-dione (**33**)

Yield 69%. White powder. M.p. 252.2 °C (decomp.). ^1^H NMR (400 MHz, CDCl_3_, δ, ppm): 1.71 (9H, s, *t*-Bu), 4.41 (3H, s, NCH_3_), 3.54 (3H, s, NCH_3_), 3.36 (3H, s, NCH_3_), 8.21 (1H, s, H-5′); ^13^C NMR (101 MHz, CDCl_3_, δ, ppm): 27.79 (NCH_3_), 29.52 (NCH_3_), 29.77 (3×CH_3_, *t*-Bu), 34.20 (NCH_3_), 60.15 (C-6′), 108.08 (C-5), 122.25 (C-5′), 138.58 (C-8), 143.09 (C-4′), 148.00 (C-4), 151.47 (C-2), 155.11 (C-6); IR (KBr, ν, cm^–1^): 1705, 1670 (C=O), 1605, 1545, 1443, 744 (C=N, C=C); UV (EtOH) λ_max_ (lgε): 230 (4.33), 299 (4.28) nm. HR-MS, *m*/*z* (I*_rel._*, %): 317 (100), 274 (14), 272 (15), 261 (28), 247 (15), 233 (25), 232 (17), 221 (13), 67 (13), 57 (12); Calcd for C_14_H_19_N_7_O_2_: 317.1595; found [M]^+^
*m*/*z:* 317.1598. Anal. calcd for C_14_H_19_N_7_O_2_, %: C, 52.99; H, 6.03; N 30.90; found, %: C, 52.72; H, 6.06; N, 28.47.

#### 3.2.5. Preparation of 8-(Piperidinyl)- or 8-(piperazinyl)-1,3,7-trimethyl-3,7-dihydro-1*H*-purine- 2,6-diones (**60**,**61**)

To a vigorously stirred solution of 8-bromocaffeine (**5**) (100 mg, 0.37 mmol) in DMF (5 mL) was added an excess (3 equiv.) of 4-methylpiperidine **46** or N-Boc-piperazine **49**. The reaction mixture was stirred at 120 °C for 3 h. After evaporation of the solvent, the residue was subjected to column chromatography (CHCl_3_-EtOH, 100:0→99:1). Compounds **60** or **61** were isolated.

##### 1,3,7-Trimethyl-8-(4-methylpiperidin-1-yl)-3,7-dihydro-1*H*-purine-2,6-dione (**60**)

Yield 78%. White powder. M.p. 131.6–131.7 °C. ^1^H NMR (500 MHz, CDCl_3_, δ, ppm): 0.97 (3H, d, J = 6.6 Hz, H-14), 1.32 (2H, quart.d, J = 12.4, 3.5 Hz, H_ax_ -12,12′), 1.49–1.61 (1H, m, H-13), 1.72 (2H, br.d, J = 12.4 Hz, H_eq_-12,12′), 2.91 (2H, td, J = 12.4, 1.6 Hz, H_ax_-11,11′), 3.34 (3H, s, NCH_3_), 3.47 (2H, br.d, J = 12.4 Hz, H_eq_ -11,11′), 3.49 (3H, s, NCH_3_), 3.68 (3H, s, N CH_3_). ^13^C NMR (126 MHz, CDCl_3_, δ, ppm): 21.68 (C-14), 27.55 (NCH_3_), 29.50 (NCH_3_), 30.43 (C-13), 32.45 (NCH_3_), 33.60 (C-12,12′), 50.06 (C-11,11′), 105.04 (C-5), 147.43 (C-4), 151.63 (C-2), 154.74 (C-8), 157.03 (C-6). IR (KBr, ν, cm^–1^): 1703, 1659 (C=O), 1612, 1520, 1452, 804, 748 (C=C, C=N); UV (EtOH) λ_max._ (lgε): 225 (4.28), 295 (4.21) nm. HR-MS, *m*/*z* (I_rel._, %): 291 (100), 276 (27), 262 (10), 248 (15), 222 (24), 221 (25), 209 (12), 208 (12), 194 (16), 69 (15); calcd for C_14_H_21_N_5_O_2_: 291.1690; found [M]^+^
*m*/*z*: 291.1689. Anal. calcd for C_14_H_21_N_5_O_2_, %: C, 57.71; H, 7.27; N 24.04; found, %: C, 58.19; H, 7.61; N, 24.37.

##### *tert*-Butyl 4-(1,3,7-trimethyl-2,6-dioxo-3,7-dihydro-1*H*-purin-8-yl)piperazine-1- carboxylate (**61**)

Yield 88%. White powder. M.p. 163.9–166.9 °C (decomp.). ^1^H NMR (500 MHz, CDCl_3_, δ, ppm): 1.45 (9H, s, *t*-Bu), 3.18 (4H, t, J = 5.0 Hz, H-11,11′), 3.49 (3H, s, NCH_3_), 3.35 (3H, s, NCH_3_), 3.56 (4H, t, J = 5.0 Hz, H-12,12′), 3.73 (3H, s, NCH_3_). ^13^C NMR (126 MHz, CDCl_3_, δ, ppm): 27.59 (NCH_3_), 28.18 (3 × CH_3_, *t*-Bu), 29.50 (NCH_3_), 32.22 (NCH_3_), 80.10 (C-12,12′), 49.43 (C-11,11′), 105.28 (C-5), 147.07 (C-4), 151.52 (C-2), 154.40 (C-8), 154.81 (C-6), 155.70 (C=O, *t*-Bu); IR (KBr, ν, cm^–1^): 1697, 1657 (C=O), 1600, 1508, 1452, 1425, 741(C=C, C=N); UV (EtOH) λ_max._ (lgε): 222 (4.29), 291 (4.20) nm. HR-MS, *m*/*z* (I*_rel_*, %): 378 (69), 322 (62), 305 (19), 278 (16), 223 (19), 222 (100), 209 (56), 67 (12), 57 (31), 41 (14); calcd for C_17_H_26_N_6_O_4_: 378.2010; found [M]^+^
*m*/*z:* 378.2011. Anal. calcd for C_17_H_26_N_6_O_4_, %: C, 53.96; H, 6.93; N, 22.21; found, %: C, 53.82; H, 6.82; N, 22.42.

#### 3.2.6. Synthesis and Spectral Data of 8-(3-Aminoprop-1-ynyl)-1,3,7-trimethyl- 3,7-dihydro-1*H*-purine-2,6-diones (**35**, **39**–**41**, **51**–**59**)

Conditions (*a*). A stirred mixture of 8-ethynylcaffeine (**24**) (218 mg, 1 mmol), 30% aq. formaldehyde (90 mg, 3 mmol), diethylamine (34) (220 mg, 3 mmol), Cu(OAc)_2_ × H_2_O (20 mg, 0.1 mmol) in THF (8 mL) was heated to 75 °C (bath) for 3 h under argon. After the consumption of the starting materials, the solvent was removed under reduced pressure, the crude material was diluted with water and the product was extracted with chloroform (3 × 15 mL). The combined organic solution was dried over MgSO_4_, filtered and evaporated in vacuo. After column chromatography (eluent CHCl_3_-EtOH, 100:0→6:1) compounds **35**,**39**–**41**,**51**–**59** were isolated. 

Conditions (*b*). Synthesis of compound (**35**) from 8-(trimethylsilylethynyl)caffeine (**26**). A stirred solution of compound (**26**) (150 mg, 0.52 mmol) in THF (5 mL) was successive treated with aq. tetra-*n*-butyl ammonium fluoride (130 mg, 0.62 mmol) for 2 h, then formalin (0.155 mL, 3 equiv. of CH_2_O), diethylamine **34** (114 mg, 1.56 mmol), and Cu(OAc)2 × H2O (10 mg, 0.05 mmol) in an argon flow. The reaction mixture was stirred 10 min at room temperature and then heated to 75 °C (bath) for 3 h and evaporated. The residue was diluted with water, and then extracted with chloroform (3 × 15 mL). The combined organic solution was dried over MgSO_4_, filtered and evaporated in vacuo. After column chromatography (eluent CHCl_3_-EtOH, 100:0→30:1) compounds **35** was isolated in the yield 44%.

##### 8-(3-(Diethylamino)prop-1-yn-1-yl)-1,3,7-trimethyl-3,7-dihydro-1*H*-purine-2,6-dione (**35**)

Yield 94%^[a]^ or 44%^[b]^. Brownish powder. M.p. 99.8–102.5 °C (decomp.). ^1^H NMR (300 MHz, CDCl_3_, δ, ppm): 1.09 (6H, t, J = 7.2 Hz, H-15,15′), 2.60 (4H, quart, J = 7.2 Hz, H-14,14′), 3.37 (3H, s, NCH_3_), 3.53 (3H, s,NCH_3_), 3.72 (2H, s, H-12), 3.97 (3H, s, NCH_3_). ^13^C NMR (126 MHz, CDCl_3_, δ, ppm): 12.59 (C-15,15′), 27.87 (N CH_3_), 29.64 (NCH_3_), 32.98 (NCH_3_), 41.21 (C-12), 47.42 (C-14,14′),72.98 (C-11), 93.73 (C-10), 107.44 (C-5), 135.37 (C-8), 147.45 (C-4), 151.38 (C-2), 154.71 (C-6). IR (KBr, ν, cm^–1^): 2183 (C≡C), 1709, 1666 (C=O), 1601, 1547, 1485, 1429, 744 (C=C, C=N); UV (EtOH) λ_max._ (lgε): 228 (4.40), 302 (4.23) nm. HR-MS, *m*/*z* (I*_rel._*, %): 302 (2), 233 (21), 233 (9), 232 (75), 231 (100), 176 (10), 175 (21), 174 (56), 146 (23), 72 (9); calcd for C_15_H_21_N_5_O_2_: 303.1690; found [M-H]^+^ *m*/*z:* 302.1607. Anal. calcd for C_15_H_21_N_5_O_2_,%: C, 59.39; H, 6.98; N, 23.09; found, %: C, 58.98; H, 6.68; N, 22.70.

##### 8-(3-(Diisopropylamino)prop-1-yn-1-yl)-1,3,7-trimethyl-3,7-dihydro-1*H*-purine- 2,6-dione (**39**)

Yield 96%. Orange powder. M.p. 113.8 °C (decomp.). ^1^H NMR (400 MHz, CDCl_3_, δ, ppm): 1.11 (12H, d, J = 6.6 Hz, 4 × CH_3_ (i-Pr)), 3.21 (2H, sept, J = 6.6 Hz, 2 × CH (i-Pr)), 3.37 (3H, s, NCH_3_), 3.53 (3H, s, NCH_3_), 3.70 (2H, s, H-12), 3.96 (3H, s, NCH_3_); ^13^C NMR (101 MHz, CDCl_3_, δ, ppm): 20.55 (4 × CH_3_, i-Pr), 27.83 (NCH_3_), 29.62 (NCH_3_), 32.84 (NCH_3_), 34.67 (C-12), 48.62 (2 × CH, i-Pr), 71.44 (C-11), 98.49 (C-10), 107.35 (C-5), 135.73 (C-8), 147.52 (C-4), 151.41 (C-2), 154.71 (C-6); IR (KBr, ν, cm^–1^): 2970, 2931, 2875, 2226 (C≡C), 1701, 1668 (C=O), 1601, 1545, 1485, 1431, 746 (C=N, C=C); UV (EtOH) λ_max_ (lgε): 229 (4.42), 302 (4.24) nm. HR-MS, *m*/*z* (I_rel_, %): 331 (4), 317 (76), 316 (37), 232 (48), 231 (100), 67(48), 57 (50), 43 (35), 42 (31), 41 (52); calcd for C_17_H_25_N_5_O_2_: 331.2003; found [M]^+^
*m*/*z*: 331.2001. Anal. calcd for C_17_H_25_N_5_O_2_,%: C, 61.61; H, 7.60; N, 21.13; found, %: C, 61.93; H, 7.35; N, 20.98.

##### 8-(3-(Dibutylamino)prop-1-yn-1-yl)-1,3,7-trimethyl-3,7-dihydro-1*H*-purine-2,6-dione (**40**)

Yield 87%. Orange powder. M.p. 65.6–67.1 °C. ^1^H NMR (300 MHz, CDCl_3_, δ, ppm): 0.90 (6H, t, J = 7.2 Hz, H-17,17′), 1.27–1.37 (4H, m, H-16,16′), 1.39–1.50 (4H, m, H-15,15′), 2.50 (4H, t, J = 7.3 Hz, H-14,14′), 3.38 (3H, s, NCH_3_), 3.54 (3H, s, NCH_3_), 3.69 (2H, s, H-12), 3.98 (3H, s, NCH_3_); ^13^C NMR (126 MHz, CDCl_3_, δ, ppm): 13.92 (C-17,17′), 20.42 (C-16,16′), 27.90 (NCH_3_), 29.60 (C-15,15′), 29.67 (NCH_3_), 32.98 (NCH_3_), 42.43 (C-12), 53.60 (C-14,14′), 72.95 (C-11), 94.29 (C-10), 107.46 (C-5), 135.48 (C-8), 147.48 (C-4), 151.42 (C-2), 154.74 (C-6); IR (KBr, ν, cm^–1^): 2229 (C≡C), 1707, 1666 (C=O), 1601, 1547, 1485, 1429, 746 (C=C, C=N); UV (EtOH) λ_max._ (lgε): 229 (4.44), 302 (4.27) nm. HR-MS, *m*/*z* (I_rel._, %): 358 (1), 316 (5), 260 (2), 233 (4), 232 (25), 231(100), 217 (2), 174 (3), 146 (2), 128 (2); calcd for C_19_H_29_N_5_O_2_: 359.2316; found [M-H]^+^
*m*/*z*: 358.2219. Anal. calcd for C_19_H_29_N_5_O_2_,%: C, 63.48; H, 8.13; N, 19.48; found, %: C, 63.82; H, 8.01; N, 19.26.

##### 8-(3-(Dicyclohexylamino)prop-1-yn-1-yl)-1,3,7-trimethyl-3,7-dihydro-1*H*-purine- 2,6-dione (**41**)

Yield 57%. Yellow powder. M.p. 147.4 °C (decomp.). ^1^H NMR (400 MHz, CDCl_3_, δ, ppm): 1.14 (2H, t.t, J = 12.2, 3.2 Hz, 2 × H_ax_-17), 1.20–1.40 (8H, m, 2 × H_ax_-15,15′,16,16′), 1.60–1.67 (2H, m, 2 × H_eq_-17), 1.74–1.83 (4H, m, 2 × H_eq_ -16,16′), 1.84–1.91 (4H, m, 2 × H_eq_-15,15′), 2.81 (2H, tt, J = 10.8, 3.3 Hz, 2 × H-14), 3.37 (3H, s, NCH_3_), 3.53 (3H, s, NCH_3_), 3.75 (2H, s, H-12), 3.96 (3H, s, NCH_3_); ^13^C NMR (101 MHz, CDCl_3_, δ, ppm): 26.01 (2 × C-16,2 × C-16′), 26.14 (2 × C-17), 28.00 (NCH_3_), 29.79 (NCH_3_), 31.36 (2 × C-15,2 × C-15′), 32.96 (NCH_3_), 35.70 (C-12), 57.53 (2 × C-14), 71.28 (C-11), 99.29 (C-10), 107.51 (C-5), 135.99 (C-8), 147.66 (C-4), 151.58 (C-2), 154.86 (C-6); IR (KBr, ν, cm^–1^): 2227 (C≡C), 1713, 1662 (C=O), 1599, 1547, 1483, 1427, 744 (C=C, C=N); UV (EtOH) λ_max._ (lgε): 229 (4.46), 302 (4.28) nm. HR-MS, *m*/*z* (I_rel._, %): 411 (33), 356 (22), 355 (100), 328 (43), 273 (24), 272 (97), 232 (15), 231 (38), 55 (32), 41 (18); calcd for C_23_H_33_N_5_O_2_: 411.2629; found [M]^+^
*m*/*z*: 411.2634. Anal. calcd for C_23_H_33_N_5_O_2_,%: C, 67.13; H, 8.08; N, 17.02; found, %: C, 66.92; H, 7.90; N 16.80.

##### 1,3,7-Trimethyl-8-(3-(pyrrolidin-1-yl)prop-1-yn-1-yl)-3,7-dihydro-1*H*-purine-2,6-dione (**51**)

Yield 62%. Orange powder. M.p. 154.1 °C (decomp.) (M.p. 149–150 °C [47]). IR (KBr, ν, cm^–1^): 2243 (C≡C), 1705, 1664 (C=O), 1599, 1547, 1487, 1429, 744 (C=O); UV (EtOH) λ_max._ (lgε): 228 (4.45), 302 (4.25) nm. ^1^H NMR, ^13^C NMR, and MS data are given in [47]. 

##### 8-(3-(Azepan-1-yl)prop-1-yn-1-yl)-1,3,7-trimethyl-3,7-dihydro-1*H*-purine-2,6-dione (**52**)

Yield 97%. Orange powder. M.p. 124.6 °C (decomp.). (M.p. 99–101 °C for hydrate of **52**) [47]). IR (KBr, ν, cm^–1^): 2239 (C≡C), 1705, 1666 (C=O), 1601, 1545, 1487, 1429, 744 (C=O); UV (EtOH) λ_max._ (lgε): 229 (4.44), 302 (4.28) nm. ^1^H NMR, ^13^C NMR, and MS data are given in [47].

##### 8-(3-(Azocan-1-yl)prop-1-yn-1-yl)-1,3,7-trimethyl-3,7-dihydro-1*H*-purine-2,6-dione (**53**)

Yield 98%. Yellow powder. M.p. 166.1–167.6 °C. ^1^H NMR (400 MHz, CDCl_3_, δ, ppm): 1.51–1.62 (10H, m, H-15,15′,16,16′,17), 2.63–2.68 (4H, m, H-14,14′), 3.36 (3H, s, NCH_3_), 3.52 (3H, s, NCH_3_), 3.64 (2H, s, H-12), 3.97 (3H, s, NCH_3_); ^13^C NMR (126 MHz, CDCl_3_, δ, ppm): 25.77 (C-16,16′), 27.35 (C-17), 27.48 (C-15,15′), 27.85 (NCH_3_), 29.62 (NCH_3_), 32.97 (NCH_3_), 48.18 (C-12), 53.11 (C-14,14′), 71.84 (C-11), 96.02 (C-10), 107.38 (C-5), 135.59 (C-8), 147.47 (C-4), 151.39 (C-2), 154.69 (C-6); IR (KBr, ν, cm^−1^): 2226 (C≡C), 1699, 1670 (C=O), 1601, 1543, 1481, 1425, 750 (C=C, C=N); UV (EtOH) λ_max._ (lgε): 229 (4.12), 302 (3.95) HM. HR-MS, *m*/*z* (I_rel._, %): 343 (21), 286 (8), 272 (9), 259 (12), 233 (13), 232 (100), 231 (68), 174 (14), 112 (88), 42 (10); calcd for C_18_H_25_N_5_O_2_: 343.2003; found [M]^+^
*m*/*z*: 343.2006. Anal. calcd for C_18_H_25_N_5_O_2_,%: C, 62.95; H, 7.34; N, 20.39; found, %: C, 63.20; H, 7.27; N, 20.28.

##### 8-(3-(1,4-Oxazepan-4-yl)prop-1-yn-1-yl)-1,3,7-trimethyl-3,7-dihydro-1*H*-purine- 2,6-dione (**54**)

Yield 90%. Orange powder. M.p. 118.7–121.7 °C. ^1^H NMR (400 MHz, CDCl_3_, δ, ppm): 1.91 (2H, pent, J = 6.0 Hz, H-18), 2.78–2.83 (4H, m, H-14,19), 3.33 (3H, s, NCH_3_), 3.50 (3H, s, NCH_3_), 3.68–3.73 (4H, m, H-12,17), 3.76 (2H, t, J = 6.2 Hz, H-15), 3.96 (3H, s, NCH_3_); ^13^C NMR (126 MHz, CDCl_3_, δ, ppm): 27.85 (NCH_3_), 29.56 (C-18), 29.61 (NCH_3_), 33.03 (NCH_3_), 48.31 (C-12), 53.24 (C-14), 57.21 (C-19), 68.44 (C-15,17), 72.92 (C-11), 93.93 (C-10), 107.47 (C-5), 135.14 (C-8), 147.41 (C-4), 151.32 (C-2), 154.66 (C-6); IR (KBr, ν, cm^–1^): 2241 (C≡C), 1705, 1662 (C=O), 1599, 1547, 1487, 1429, 742 (C=C, C=N); UV (EtOH, λ_max_, nm) (lgε): 228 (4.44), 302 (4.26). HR-MS, *m*/*z* (I_rel._, %): 331 (100), 272 (23), 233 (16), 232 (95), 231 (58), 194 (39), 100 (24), 82 (19), 67 (31), 42 (38); calcd for C_16_H_21_N_5_O_3_: 331.1639; found [M]^+^
*m*/*z*: 331.1642. Anal. calcd for C_16_H_21_N_5_O_3_, %: C, 57.99; H, 6.39; N, 21.13; found, %: C, 57.25; H, 6.35; N 20.67.

##### 1,3,7-Trimethyl-8-(3-(4-methylpiperidin-1-yl)prop-1-yn-1-yl)-3,7-dihydro-1*H*- purine-2,6-dione (**55**)

Yield 91%. Yellow powder. M.p. 123.0–125.8 °C. ^1^H NMR (500 MHz, CDCl_3_, δ, ppm): 0.93 (3H, d, J = 5.9 Hz, CH_3_-17), 1.26–1.39 (3H, m, H_ax_ -14,14′,16), 1.68 (2H, d, J = 11.6 Hz, H_eq_-14,14′), 2.32 (2H, dd, J = 11.3, 12.8 Hz, H_ax_-15,15′), 2.93 (2H, d, J = 11.3, H_eq_ -15,15′), 3.38 (3H, s, NCH_3_), 3.54 (3H, s, NCH_3_), 3.64 (2H, s, H-12), 4.00 (3H, s, NCH_3_); ^13^C NMR (126 MHz, CDCl_3_, δ, ppm): 21.65 (C-17), 27.90 (NCH_3_), 29.66 (NCH_3_), 29.98 (C-16), 33.07 (NCH_3_), 34.01 (C-15,15′), 47.81 (C-12), 52.74 (C-14,14′), 73.12 (C-11), 94.02 (C-10), 107.49 (C-5), 135.31 (C-8), 147.46 (C-4), 151.39 (C-2), 154.73 (C-6); IR (KBr, ν, cm^−1^): 2245 (C≡C), 1707, 1668 (C=O), 1601, 1545, 1487, 1429, 744 (C=C, C=N); UV (EtOH) λ_max._ HM (lgε): 229 (4.45), 302 (4.27). HR-MS, *m*/*z* (I_rel._, %): 328(16), 233 (17), 232 (100), 231 (43), 112 (20), 98 (56), 67 (21), 55 (16), 42 (21), 41 (18); calcd for C_17_H_23_N_5_O_2_: 329.1846; found [M-H]^+^
*m*/*z*: 328.1762. Anal. calcd for C_17_H_23_N_5_O_2_, %: C, 61.99; H, 7.04; N, 21.26; found, %: C, 61.52; H, 6.85; N, 20.92.

##### 1,3,7-Trimethyl-8-(3-morpholinoprop-1-yn-1-yl)-3,7-dihydro-1*H*-purine-2,6-dione (**56**)

Yield 89%. White powder. M.p. 184.3–186.0 °C.). (M.p. 88–190 °C [47]). ^1^H NMR (400 MHz, CDCl_3_, δ, ppm): 2.60 (4H, t, J = 4.5 Hz, H-14,14′), 3.35 (3H, s, NCH_3_), 3.51 (3H, s, NCH_3_), 3.60 (2H, s, H-12), 3.71 (4H, t, J = 4.5 Hz, H-15,15′), 3.97 (3H, s, NCH_3_); ^13^C NMR (126 MHz, CDCl_3_, δ, ppm): 27.86 (NCH_3_), 29.60 (NCH_3_), 33.05 (NCH_3_), 47.66 (C-12), 52.07 (C-14,14′), 66.58 (C-15,15′), 73.53 (C-11), 92.96 (C-10), 107.51 (C-5), 135.00 (C-8), 147.42 (C-4), 151.32 (C-2), 154.68 (C-6); IR (KBr, ν, cm^–1^): 2226 (C≡C), 1699, 1664 (C=O), 1545, 1487, 1429, 746 (C=C, C=N); UV (EtOH) λ_max._ (lgε): 228 (4.43), 302 (4.26) nm. HR-MS, *m*/*z* (I_rel._, %): 317 (20), 232 (100), 231 (31), 86 (21), 67 (40), 56 (46), 46 (23), 45 (48), 43 (26), 42 (38); calcd for C_15_H_19_N_5_O_3_: 317.1482; found [M]^+^
*m*/*z*: 317.1486. Anal. calcd for C_15_H_19_N_5_O_3_, %: C, 56.77; H, 6.03; N, 22.07; found, %: C, 56.42; H, 5.97; N, 22.00.

##### 1,3,7-Trimethyl-8-(3-(4-methylpiperazin-1-yl)prop-1-yn-1-yl)-3,7-dihydro-1*H*- purine-2,6-dione (**57**)

Yield 53%. Orange powder. M.p. 144.1 °C (decomp.). ^1^H NMR (400 MHz, CDCl_3_, δ, ppm): 2.25 (3H, s, CH_3_-17), 2.38–2.54 (4H, m, H-15,15′), 2.60–2.69 (4H, m, H-14,14′), 3.34 (3H, s, NCH_3_), 3.50 (3H, s, NCH_3_), 3.60 (2H, s, H-12), 3.95 (3H, s, NCH_3_); ^13^C NMR (126 MHz, CDCl_3_, δ, ppm): 27.85 (NCH_3_), 29.59 (NCH_3_), 30.07 (NCH_3_), 45.78 (C-17), 47.29 (C-12), 51.71 (C-14,14′), 54.69 (C-15,15′), 73.42 (C-11), 93.30 (C-10), 107.45 (C-5), 135.13 (C-8), 147.42 (C-4), 151.34 (C-2), 154.68 (C-6); IR (KBr, ν, cm^–1^): 2243 (C≡C), 1705 1666 (C=O), 1599, 1547, 1487, 1427, 744 (C=C, C=N); UV (EtOH) λ_max._ (lgε): 228 (4.44), 302 (4.26) nm. HR-MS, m/z (I_rel._, %): 330 (57), 317 (50), 70 (49), 67 (57), 57 (100), 56 (94), 55 (49), 43 (76), 42 (92), 41 (81); calcd for C_16_H_22_N_6_O_2_: 330.1799; found [M]^+^
*m*/*z*: 330.1795. Anal. calcd for C_16_H_22_N_6_O_2_, %: C, 58.17; H, 6.71; N, 25.44; found, %: C, 57.87; H, 6.56; N, 25.81. 3.2.6.12. tert-Butyl 4-(3-(1,3,7-trimethyl-2,6-dioxo-3,7-dihydro-1H-purin-8-yl)prop- 2-yn-1-yl)piperazine-1-carboxylate (58). Yield 58%. Yellow powder. M.p. 151.0–152.0 °C. ^1^H NMR (400 MHz, CDCl_3_, δ, ppm): 1.42 (9H, s, t-Bu), 2.55 (4H, t, J = 4.8, H-14,14′), 3.36 (3H, s, NCH_3_), 3.45 (4H, t, J = 4.8 Hz, H-15,15′), 3.52 (3H, s, NCH_3_), 3.63 (2H, s, H-12), 3.97 (3H, s, NCH_3_); ^13^C NMR (101 MHz, CDCl_3_, δ, ppm): 27.87 (NCH_3_), 28.24 (3 × CH_3_, t-Bu), 29.62 (NCH_3_), 33.05 (NCH_3_), 47.51 (C-12), 51.61 (C-14,14′,15,15′), 73.59 (C-11), 79.76 (C, t-Bu), 92.93 (C-10), 107.56 (C-5), 135.02 (C-8), 147.46 (C-4), 151.36 (C-2), 154.47 (C=O, t-Bu), 154.71 (C-6); IR (KBr, ν, cm^–1^): 2231 (C≡C), 1705, 1672 (C=O), 1545, 1487, 1423, 742(C=C, C=N); UV (EtOH) λ_max._ (lgε): 229 (4.44), 302 (4.28) nm. HR-MS, *m*/*z* (I_rel._, %): 416 (14), 343 (17), 316 (33), 315 (27), 232 (80), 231 (52), 85 (12), 84 (14), 57 (100), 41 (17); calcd for C_20_H_28_N_6_O_4_: 416.2167; found [M]^+^
*m*/*z*: 416.2172. Anal. calcd for C_20_H_28_N_6_O_4_, %: C, 57.68; H, 6.78; N, 20.18; found, %: C, 57.74; H, 6.82; N, 20.21.

##### 1,3,7-Trimethyl-8-(3-(4-(2-(pyrrolidin-1-yl)ethyl)piperazin-1-yl)prop-1-yn-1-yl) -3,7-dihydro-1*H*-purine-2,6-dione hydrate (**59**)

Yield 75%. Yellow powder. M.p. 208.4 °C (decomp.). ^1^H NMR (300 MHz, CDCl_3_, δ, ppm): 2.00–2.09 (4H, m, H-21,21′), 2.51–2.59 (4H, m, H-15,15′), 2.60–2.68 (4H, m, H-14,14′), 2.83 (2H, br.t, J = 6.3 Hz, H-17), 3.05 (2H, br.t, J = 6.3 Hz, H-18), 3.17–3.27 (4H, m, H-20,20′), 3.34 (3H, s, NCH_3_), 3.50 (3H, s, NCH_3_), 3.58 (2H, s, H-12), 3.96 (s, 3H, NCH_3_); ^13^C NMR (126 MHz, CDCl_3_, δ, ppm): 23.08 (C-21,21′), 27.90 (NCH_3_), 29.64 (NCH_3_), 33.14 (NCH_3_), 47.25 (C-12), 51.76 (C-14,14′), 52.01 (C-18), 52.78 (C-15,15′), 53.29 (C-17), 54.15 (C-20,20′), 73.35 (C-11), 93.24 (C-10), 107.56 (C-5), 135.06 (C-8), 147.42 (C-4), 151.36 (C-2), 154.69 (C-6); IR (KBr, ν, cm^–1^): 2226 (C≡C), 1701, 1672 (C=O), 1545,1421, 1340, 748 (C=C, C=N); UV (EtOH) λ_max._ (lgε): 228 (4.41), 302 (4.24) nm. HR-MS, *m*/*z* (I_rel._, %): 413 (2), 329 (26), 231 (11), 85 (12), 84 (100), 83 (18), 47 (9), 42 (13), 36 (8), 18 (13); calcd for C_21_H_31_N_7_O_2_: 413.2534; found [M]^+^
*m*/*z*: 413.2535. Anal. calcd for C_21_H_31_N_7_O_2_ × H_2_O, %: C, 58.47; H, 7.65; N 22.73; found, %: C, 58.13; H, 7.48; N, 22.29.

#### 3.2.7. Propargylation of Dimethylxanthines (**2**,**3**)

General Procedure

This procedure is a modification of the method used by Daly et al. [59]. To a stirred suspension of dialkylxanthine **2** or **3** (1000 mg, 5.6 mmol) and anhydrous K_2_C0_3_ (767 mg, 5.6 mmol) in DMF (30 mL) was added dropwise propargyl bromide **63** (660 mg, 5.6 mmol). The reaction mixture was stirred at rt for 24 h, and the volatile material was evaporated on a Petri dish at ambient temperature. The residue was treated with H_2_O and extracted with several portions of CHCl_3_. The combined extract was dried over MgSO_4_, filtered, and the solvent was evaporated in vacuo. The obtained compounds **62** or **67** were used without further purification.

#### 3.2.8. A^3^-Coupling of 1-Prop-2-ynyl-, or 7-Prop-2-ynyl- methylxanthines. Synthesis and Spectral Data of 1-(4-Aminobut-2-yn-1-yl)-3,7-dimethyl-3,7-dihydro-1*H*-purine-2,6-diones (**64**–**66**) or 7-(4-Aminobut-2-yn-1-yl)-1,3-dimethyl-3,7-dihydro-1*H*-purine-2,6-diones (**68**–**70**)

A stirred mixture of 1-prop-2-ynyl-, or 7-prop-2-ynylmethylxanthines (**62**, **67**) (150 mg, 0.7 mmol), 30% aq. formaldehyde (63 mg, 2.1 mmol), secondary amine (2.1 mmol), Cu(OAc)_2_ × H_2_O (14 mg, 0.07 mmol) in THF (5 mL) was heated to 75 °C (bath) for 3 h under argon. After the consumption of the starting materials, the solvent was removed under reduced pressure, the crude material was diluted with water and the product was extracted with chloroform (3 × 15 mL). The combined organic solution was dried over MgSO_4_, filtered, and evaporated in vacuo. After column chromatography (eluent CHCl_3_-EtOH, 100:0→6:1), compounds **64**–**66** or **68**–**70** were isolated. 

##### 1-(4-(Diisopropylamino)but-2-yn-1-yl)-3,7-dimethyl-3,7-dihydro-1*H*-purine-2,6-dione (**64**)

Yield 90%. White powder. M.p. 46.5 °C (decomp.). ^1^H NMR (400 MHz, CDCl_3_, δ, ppm): 1.00 (12H, d, J = 6.5 Hz, 4 × CH_3_ (i-Pr)), 3.13 (2H, sept, J = 6.5 Hz, 2 × CH (i-Pr)), 3.34 (2H, t, J = 1.9 Hz, H-13), 3.49 (3H, s, NCH_3_), 3.91 (s, 3H, NCH_3_), 4.68 (2H, t, J = 1.9 Hz, H-10), 7.48 (1H, s, H-8); ^13^C NMR (101 MHz, CDCl_3_, δ, ppm): 20.05 (4 × CH_3_, i-Pr), 29.48 (NCH_3_), 30.64 (C-10), 33.38 (NCH_3_), 34.06 (C-13), 48.42 (2 × CH, i-Pr), 78.38 (C-12), 80.22 (C-11), 107.23 (C-5), 141.43 (C-8), 148.69 (C-4), 150.58 (C-2), 154.08 (C-6). HR-MS, *m*/*z* (I_rel._, %): 331 (11), 317 (20), 316 (95), 274 (5), 232 (16), 231 (100), 229 (3), 181 (4), 136 (3), 94 (3); calcd for C_17_H_25_N_5_O_2_: 331.2003; found [M]^+^
*m*/*z*: 331.2006. Anal. calcd for C_17_H_25_N_5_O_2_, %: C, 61.61; H, 7.60; N, 21.13; found, %: C, 61.23; H, 7.37; N, 20.68.

##### 1-(4-(Azocan-1-yl)but-2-yn-1-yl)-3,7-dimethyl-3,7-dihydro-1*H*-purine-2,6-dione (**65**)

Yield 83%. Yellow powder. M.p. 77.5°C (decomp.). ^1^H NMR (400 MHz, CDCl_3_, δ, ppm): 1.40–1.53 (10H, m, H-16,16′,17,17′,18), 2.46–2.53 (4H, m, H-15,15′), 3.22 (2H, t, J = 1.8 Hz, H-13), 3.49 (3H, s, NCH_3_),3.91 (3H, s, NCH_3_), 4.70 (2H, t, J = 1.8 Hz, H-10), 7.47 (1H, s, H-8); ^13^C NMR (101 MHz, CDCl_3_, δ, ppm): 25.76 (C-17,17′), 27.17 (C-18), 27.26 (C-16,16′), 29.48 (NCH_3_), 30.53 (C-10), 33.35 (NCH_3_), 47.53 (C-13), 52.78 (C-15,15′), 77.97 (C-12), 78.84 (C-11), 107.25 (C-5), 141.41 (C-8),148.68 (C-4), 150.58 (C-2), 154.09 (C-6). HR-MS, *m*/*z* (I_rel._, %): 343 (26), 231 (80), 150 (40), 112 (100), 107 (21), 67 (28), 55 (32), 42 (48), 41 (40), 18 (32); calcd for C_18_H_25_N_5_O_2_: 343.2003; found [M]^+^
*m*/*z*: 343.2002. Anal. calcd for C_18_H_25_N_5_O_2_, %: C, 62.95; H, 7.34; N, 20.39; found, %: C, 62.61; H, 7.28; N, 19.86.

##### 3,7-Dimethyl-1-(4-(4-(2-(pyrrolidin-1-yl)ethyl)piperazin-1-yl)but-2-yn-1-yl)-3,7- dihydro-1*H*-purine-2,6-dione hydrate (**66**)

Yield 54%. Orange amorphous substance. M.p. not determ. ^1^H NMR (400 MHz, CDCl_3_, δ, ppm): 7.47 (1H, s, H-8), 4.72 (2H, t, J = 1.8 Hz, H-10), 3.92 (3H, s,NCH_3_), 3.51 (3H, s, NCH_3_), 3.17 (2H, t, J = 1.8 Hz, H-13), 2.42–2.59 (16H, m, H-15,15′,16,16′,18,19,21,21′), 1.68–1.73 (4H, m,H-22, 22′); ^13^C NMR (101 MHz, CDCl_3_, δ, ppm): 154.13 (C-6), 150.63 (C-2), 148.76 (C-4), 141.46 (C-8), 107.32 (C-5), 79.59 (C-11), 76.98 (C-12), 57.18 (C-18), 54.28 (C-21, C-21′), 53.36 (C-19), 53.21 (C-16, C-16′), 51.77 (C-15, C-15′), 46.93 (C-13), 33.41 (NCH_3_), 30.52 (C-10), 29.55 (NCH_3_), 23.14 (C-22,22′). HR-MS, *m*/*z* (I_rel_, %): 413 (24), 330 (18), 329 (100), 317 (8), 316 (8), 231 (13), 180 (7), 149 (34), 84 (31); calcd for C_21_H_31_N_7_O_2_: 413.2534; found [M]^+^
*m*/*z*: 413.2532. Anal. calcd for C_21_H_31_N_7_O_2_ × H_2_O, %: C, 58.47; H, 7.65; N 22.73; found, %: C, 58.13; H, 7.48; N, 22.29.

##### 7-(4-(Diisopropylamino)but-2-yn-1-yl)-1,3-dimethyl-3,7-dihydro-1*H*-purine-2,6-dione (**68**)

Yield 98%. White powder. M.p. 63.7–65.3 °C. ^1^H NMR (400 MHz, CDCl_3_, δ, ppm): 1.00 (12H, d, J = 6.6 Hz, 4 × CH_3_ (i-Pr)), 3.08 (2H, sept, J = 6.6 Hz, 2 × CH (i-Pr)), 3.31 (3H, s, NCH_3_), 3.39 (2H, t, J = 1.9 Hz, H-13), 3.50 (3H, s, NCH_3_), 5.07 (2H, t, J = 1.8 Hz, H-10), 7.77 (s, 1H, H-8); ^13^C NMR (126 MHz, CDCl_3_, δ, ppm): 20.27 (4 × CH_3_, i-Pr), 27.69 (NCH_3_), 29.52 (NCH_3_), 34.04 (C-13), 36.91 (C-10), 48.25 (2 × CH, i-Pr), 74.08 (C-12), 87.56 (C-11), 106.44 (C-5), 140.15 (C-8), 148.55 (C-4),151.32 (C-2), 154.90 (C-6). HR-MS, *m*/*z* (I_rel_, %): 331 (8), 317 (22), 316 (100), 232 (19), 231 (21), 152 (29), 138 (36), 94 (26), 43 (26), 41 (17); calcd for C_17_H_25_N_5_O_2_: 331.2000; found [M]^+^
*m*/*z*: 331.2003. Anal. calcd for C_17_H_25_N_5_O_2_, %: C, 61.61; H, 7.60; N, 21.13; found, %: C, 61.68; H, 7.77; N, 21.21.

##### 7-(4-(Azocan-1-yl)but-2-yn-1-yl)-1,3-dimethyl-3,7-dihydro-1*H*-purine-2,6-dione (**69**)

Yield 84%. Yellow powder. M.p. 46.9 °C (decomp.). ^1^H NMR (400 MHz, CDCl_3_, δ, ppm): 7.85 (1H, s, H-8), 5.15 (2H, t, J = 1.7 Hz, H-10), 3.57 (3H, s, NCH_3_), 3.40 (2H, t, J = 1.9 Hz, H-13), 3.38 (3H, s, NCH_3_), 2.56–2.61 (4H, m, H-15,15′), 1.51–1.60 (10H, m, H-16,16′,17,17′,18). ^13^C NMR (126 MHz, CDCl_3_, δ, ppm): 25.83 (C-17,17′), 27.27 (C-18), 27.42 (C-16,16′), 27.84 (NCH_3_), 29.66 (NCH_3_), 36.92 (C-10), 47.59 (C-13), 95.43 (C-11), 74.71 (C-12), 53.11 (C-15,15′), 106.63 (C-5), 140.30 (C-8), 148.76 (C-4), 151.50 (C-2), 155.10 (C-6). HR-MS, *m*/*z* (I_rel._, %): 343 (1), 342 (18), 231 (24), 180 (37), 164 (63), 163 (37), 150 (100), 112 (56), 55 (28), 41 (33); calcd for C_18_H_25_N_5_O_2_: 343.2003; found [M-H]^+^*m*/*z*: 342.1922. Anal.calcd for C_18_H_25_N_5_O_2_,%: C, 62.95; H, 7.34; N, 20.39; found,%: C, 62.65; H, 7.27; N, 19.91.

##### 1,3-Dimethyl-7-(4-(4-(2-(pyrrolidin-1-yl)ethyl)piperazin-1-yl)but-2-yn-1-yl)-3,7- dihydro-1*H*-purine-2,6-dione (**70**)

Yield 74%. White amorphous powder; mp not determ. ^1^H NMR (300 MHz, CDCl_3_, δ, ppm): 1.76–1.84 (4H, m, H-22,22′), 2.49–2.71 (16H, m, H-15,15′,16,16′,18,19,21,21′), 3.32 (2H, t, J = 1.9 Hz, H-13), 3.37 (3H, s, NCH_3_), 3.56 (3H, s, NCH_3_), 5.15 (2H, t, J = 1.7 Hz, H-10), 7.80 (1H, s, H-8); ^13^C NMR (75 MHz, CDCl_3_, δ, ppm): 23.22 (C-22,22′), 27.83 (NCH_3_), 29.66 (NCH_3_), 36.69 (C-10), 46.88 (C-13), 51.91 (C-15,15′), 53.17 (C-16,16′), 53.29 (C-19), 54.37 (C-21,21′), 56.72 (C-18), 76.59 (C-12), 83.11 (C-11), 106.59 (C-5), 140.36 (C-8), 148.72 (C-4), 151.49 (C-2), 155.08 (C-6). HR-MS, m/z (I_rel._, %): 413 (1), 127 (4), 85 (5), 84 (100), 70 (4), 56 (7), 55 (5), 42 (12), 41 (4), 28 (3); calcd for C_21_H_31_N_7_O_2_: 413.2534; found [M]^+^
*m*/*z*: 413.2532. Anal. calcd for C_21_H_31_N_7_O_2_, %: C, 61.00; H, 7.56; N, 23.71; found, %: C, 60.61; H, 7.40; N, 23.31.

### 3.3. Biochemical Method

AChE activity inhibition was evaluated by the assay described by Ellman et al. [52]. 5,5′-dithiobis(2-nitrobenzoic acid) (DTNB), AChE (AChE, E.C.3.1.1.7, Type V-S, lyophilized powder, from electric eel, 1000 units), and acetylthiocholine iodide were obtained from Sigma. A total of 100 μL of a solution containing 10 mM Tris-HCl pH7.5, 0.5 mM DTNB, and 1 μL of acetylcholinesterase at a concentration of 6.25 units × 10^−3^/mL was dropped into polystyrene wells in each. Then, 2 μL of solutions of inhibitors in DMSO were added. The control was supplemented with 2 µL of DMSO without inhibitors. Incubation was performed at room temperature for 15 min., then 5 min. at 4 °C on a planetary shaker. After that, 100 µL of a cooled solution containing 10 mM Tris-HCl pH7.5, 0.5 mM DTNB, and 1 mM acetylthiocholine iodide, prepared immediately before addition, was added to each well. The wells were placed in a reader (PerkinElmer 2103 Multilabel Reader), and the reaction was monitored at room temperature at a wavelength of 405 nm. Each inhibitor was applied at three concentrations in half-log increments (eg 10, 30, and 100) in triplicate. The measurements were carried out at a wavelength of 405 nm every 2 min. The 10th measurement was used for calculations (18 min. from the beginning of the reaction). For each well, the first measurement (background) was subtracted from the indicator of the last measurement, and the resulting numbers were used for further calculations.

Next, activity curves were built, where the point without inhibitor was taken as 100%. The points were used to build a trend line from which the inhibitor concentration where the activity was at 50% was calculated All constructions and determination of the mean and standard deviation were made in Excel 2007 and online services for solving equations. The curves for AChE inhibition by compounds **28**, **64**, **65**, **66**, and **70** are given in Appendix A.

### 3.4. Molecular Modelling and Molecular Dynamic Procedures

Molecular modeling was carried out in the Schrodinger Maestro visualization environment using applications from the *Schrodinger Small Molecule Drug Discovery Suite 2016-1* package [60]. Three-dimensional structures of the derivatives were obtained empirically in the LigPrep application using the OPLS3 force field [61]. All possible tautomeric forms of compounds, as well as various states of polar protons of molecules in the pH range of 7.0 ± 2.0 were taken into account. For calculations, the XRD model of human AChE with PDB ID 6O4W (resolution 2.35 Å) was chosen [62]. To model a possible mechanism of inhibition of the selected target, molecular docking of new compounds was performed at the binding site of donepezil of AChE in the Glide application [63]. The search area for docking was selected automatically, based on the size and physico-chemical properties of the inhibitor. The extra precision (XP) algorithm of docking was applied. Docking was performed in comparison with donepezil. The molecular structure of the inhibitor was obtained in the PubChem database and prepared in the LigPrep application. Non-covalent interactions of compounds in the binding site were visualized using Biovia Discovery Studio Client [64]. 

The molecular dynamic simulation was performed by using NAMD v. 2.14 [65]. Topology files for the ligands were generated by using SwissParam [66], and simulations were performed by using the CHARMM force field [67]. The protein–ligand complexes were solvated using a cubic box with periodic boundary conditions, with the minimal distance from the protein to the boundary being 20 Å. Water molecules, located in the protein pocket and taken into the account during docking procedure, were added as well into the systems to be simulated. The systems were neutralized by the addition of sodium or chlorine ions. Simulations were performed in conditions of 0.15 M NaCl solution. The systems were minimized, annealed up to 310 K, and equilibrated during 1 nanosecond in an NPT ensemble with the movable of the backbone being constrained. The following simulations were performed during 100 nanoseconds in an NVT ensemble at 310 K. System generations and calculations of the ligand root-mean-square deviations (RMSD) were performed in VMD [68]. Plots of changes in the RMSD of atomic coordinates over simulation time, obtained as a result of molecular dynamics research, are given in Appendix A.

## 4. Conclusions

We have proposed A^3^-coupling approaches for combining methylxanthine scaffold with nitrogen-modified 8-(3-aminoprop-1-ynyl)-, 1-aminobut-2-ynyl-, and 7-aminobut-2-ynyl- substituent to prepare new candidate molecules as AChE inhibitors. Among these synthesized compounds, 8-(3-(azocan-1-yl)prop-1-ynyl)- **53**, 8-(3-(4-(2-(pyrrolidin-1-yl)- ethyl)-piperazin-1-yl)prop-1-ynyl)caffeines **59**, 1-(4-(azocan-1-yl)but-2-ynyl)-1-(4-(4- (2-(pyrrolidin-1-yl)ethyl)piperazin-1-yl)but-2-ynyl)theobromines **65**, **66**, and 7-(4- (azocan-1-yl)but-2-ynyl)theophylline **69** exhibited the AChE inhibitory potency in nanomolar concentrations. Molecular docking studies analyzed the binding mode of the most active molecules toward AChE. The majority of the active compounds interacted with the residues of Tyr341, Phe338, Trp86, and Phe295 amino acids, important residues of the AChE binding pocket, indicating that all ligands fitted well in the catalytic site, and might be an explanation for the relevant in vitro activities. In sum, some of the described methylxanthine derivatives showed promising properties for further study. In addition, due to the simplicity and effectiveness of the reported A^3^-coupling methods, it is likely that the new transformation of methylxanthine core will find use in the development of compounds that could be used as scaffolds toward accessing other libraries of bioactive compounds.

## Data Availability

Not applicable.

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
