# Peer review of "Design, Synthesis and Assay of Novel Methylxanthine–Alkynylmethylamine Derivatives as Acetylcholinesterase Inhibitors"

_molecules, 2022, doi:10.3390/molecules27248787_

Round 1
Reviewer 1 Report
The manuscript entitled “Design, Synthesis and Assay Of Novel Methylxanthine – Alkynylmethylamine Derivatives as Acetylcholinesterase Inhibitors” aims to describe the design and synthesis of 38 methylxanthine derivatives and test them as acetylcholinesterase inhibitors. The bioactivities of all new compounds were evaluated by Ellman’s method, and the results showed that most of the synthesized compounds displayed good and moderate acetylcholinesterase inhibitory activities in vitro. The structure-activity relationships were also discussed. This topic is highly important since we witness the epidemic of Alzheimer’s disease worldwide nowadays. The manuscript is very well prepared. The results are highly valuable and well-presented. The Discussion is good.
I have some minor comments:
Figure 1: Please, label 1-3 compounds with the names.
I suggest that authors comment on compounds with IC50 below the galantamine value and those with comparable values. Is it better to have a strong or moderate AChE inhibitor? It should be analyzed with the potential side effects of future drugs in mind. See https://pubmed.ncbi.nlm.nih.gov/33999717/.
Author Response
Authors were very grateful for the valuable remarks from Referee 1. We have worked through all comments carefully and made the corresponding changes to the manuscript for better presentation of our results.
1) Figure 1: Please, label 1-3 compounds with the names.
Figure 1 was corrected.
2) I suggest that authors comment on compounds with IC50 below the galantamine value and those with comparable values. Is it better to have a strong or moderate AChE inhibitor? It should be analyzed with the potential side effects of future drugs in mind. See https://pubmed.ncbi.nlm.nih.gov/33999717/.
Our Reviewer raises a very important debatable question. At this moment we don’t know what is better to have a strong or moderate AChE inhibitor. Initially, at this stage of our work, we showed that methylxanthines with propargylamine or 4-(amino)but-2-yn-1-yl) substituent could be considered as novel valuable lead compounds in the design of future drugs. The prospect of searching for acetylcholinesterase inhibitors in the methylxanthine series is due to the recent findings about caffeine´s impact on neurodegenerative diseases [References 13-16]. An inspiring example in the direction of the development of our research was the data [53] about propargyl dimethylxanthines as potent and selective in vivo antagonists of adenosine receptors. Thank you very much to the Referee for the reference to a very important and excellent work, which will provide new insights into the SARs and binding interactions of this type of AChE inhibitors, and will be very useful for the future in vitro and in vivo studies of this novel groups of compounds.
We made important corrections and additions to the manuscript, which were necessary for the better presentation of our scientific material. Thank you very much for all the comment.
Yours sincerely,
Elvira Shults

Reviewer 2 Report
1. in the abstact lack of precise numbers from the research results.
2. page 6 - 1H NMR for 5a compound has error
3. Each figure/chart requires SD and desription of the axes with units
4. no description of the performance of ellman's assay, kinetic assay and beta-amyloid assay. No literature references.
5. Please provide exact numbers for the ellman's assay and how IC50 was calculated.
Author Response
Authors were very grateful for the valuable remarks from Referee. We have worked through all comments carefully and made the corresponding changes to the manuscript for better presentation of our results.
1) We have added in the Abstract precise numbers of IC50 0.25, 0.552, 0.089, 0.746 and 0.121 μM for cоmpounds 53, 59, 65, 66, 69. Additionally the yield of the A3-coupling reactions was also added in the Abstract ((yield 53-96%).
2) This error was corrected.
3) The Suppl. part was corrected
4) and 5)
AChE activity inhibition was evaluated by the assay described by Ellman et al. [52]. 5,5′-dithiobis(2-nitrobenzoic acid) (DTNB), AChE (AChE, E.C.3.1.1.7, Type V-S, lyophilized powder, from electric eel, 1000 units), acetylthiocholine iodide were obtained from Sigma. 100 μL of a solution containing 10 mM Tris-HCl pH7.5, 0.5 mM DTNB, and 1 μL of acetylcholinesterase at a concentration of 6.25 units ×10-3/mL were dropped into polystyrene wells in each. Then, 2 μL of solutions of inhibitors in DMSO were added. The control was supplemented with 2 µL of DMSO without inhibitors. Incubated was done at room temperature for 15 min., then 5 min. at 4oC on a planetary shaker. After that, 100 µL of a cooled solution containing 10 mM Tris-HCl pH7.5, 0.5 mM DTNB, 1 mM acetylthiocholine iodide, prepared immediately before addition, was added to each well. The wells were placed in a reader (PerkinElmer 2103 Multilabel Reader) and the reaction was monitored at room temperature at a wavelength of 405 nm. Each inhibitor was applied at three concentrations in half log increments (eg 10, 30 and 100) in triplicate. The measurements were carried out at a wavelength of 405 nm every 2 min. The 10th measurement was used for calculations (18 min. from the beginning of the reaction). For each well, the first measurement (background) was subtracted from the indicator of the last measurement, the resulting numbers were used for further calculations.
Next, activity curves were built, where the point without inhibitor was taken as 100%. The points were used to build a trend line from which the inhibitor concentration was calculated at which the activity was 50%. All constructions, determination of the mean and standard deviation were made in Excel 2007 and online services for solving equations. The curves for AChE inhibition by compounds 28, 64, 65, 66, 70 were added and given in Supplementary part (Fig S5A-S5E).
Thank you very much for all the comment. All errors have been corrected. We made important corrections and additions to the manuscript, which were necessary for the better presentation of our scientific material. We appreciate of the helpful suggestions from Referee 2.
Yours sincerely,
Elvira Shults
